# Tyramine and its *Amtyr1* receptor modulate attention in honey bees (*Apis mellifera*)

Joseph S Latshaw, Reece E Mazade, Mary Petersen, Julie A Mustard[†], Irina Sinakevitch[‡], Lothar Wissler, Xiaojiao Guo, Chelsea Cook, Hong Lei, Jürgen Gadau[§], Brian Smith*

School of Life Sciences, Arizona State University, Tempe, United States

*For correspondence:
brian.h.smith@asu.edu

Present address: [†]School of Integrative Biological and Chemical Sciences, University of Texas Rio Grande Valley, Brownsville, United States; [‡]Evelyn F. McKnight Brain Institute, University of Arizona, Tucson, United States; [§]Institute for Evolution und Biodiversity, University of Münster, Münster, Germany

Competing interest: The authors declare that no competing interests exist.

**Abstract** Animals must learn to ignore stimuli that are irrelevant to survival and attend to ones that enhance survival. When a stimulus regularly fails to be associated with an important consequence, subsequent excitatory learning about that stimulus can be delayed, which is a form of nonassociative conditioning called 'latent inhibition'. Honey bees show latent inhibition toward an odor they have experienced without association with food reinforcement. Moreover, individual honey bees from the same colony differ in the degree to which they show latent inhibition, and these individual differences have a genetic basis. To investigate the mechanisms that underly individual differences in latent inhibition, we selected two honey bee lines for high and low latent inhibition, respectively. We crossed those lines and mapped a Quantitative Trait Locus for latent inhibition to a region of the genome that contains the tyramine receptor gene *Amtyr1* [We use Amtyr1 to denote the gene and AmTYR1 the receptor throughout the text.]. We then show that disruption of *Amtyr1* signaling either pharmacologically or through RNAi qualitatively changes the expression of latent inhibition but has little or slight effects on appetitive conditioning, and these results suggest that AmTYR1 modulates inhibitory processing in the CNS. Electrophysiological recordings from the brain during pharmacological blockade are consistent with a model that AmTYR1 indirectly regulates at inhibitory synapses in the CNS. Our results therefore identify a distinct *Amtyr1*-based modulatory pathway for this type of nonassociative learning, and we propose a model for how *Amtyr1* acts as a gain control to modulate hebbian plasticity at defined synapses in the CNS. We have shown elsewhere how this modulation also underlies potentially adaptive intracolonial learning differences among individuals that benefit colony survival. Finally, our neural model suggests a mechanism for the broad pleiotropy this gene has on several different behaviors.

## Editor's evaluation

This article reports a significant discovery: disrupting the function of the tyramine receptor in honey bees causes a rapid decline in their responses to olfactory stimuli. This finding highlights the important role of tyramine receptors, one of the most highly expressed biogenic amine receptors in the insect olfactory system. The authors propose that tyramine signaling may specifically control the process of latent inhibition, but the evidence presented does not rule out the possibility that tyramine affects other functions of the antennal lobe.

## Introduction

The ability to learn predictive associations between stimuli and important events, such as food or threats, is ubiquitous among animals (*Heyes, 2012*), and it may underlie more complex cognitive

**eLife digest** To efficiently navigate their environment, animals must pay attention to cues associated with events important for survival while also dismissing meaningless signals. The difference between relevant and irrelevant stimuli is learned through a range of complex mechanisms that includes latent inhibition. This process allows animals to ignore irrelevant stimuli, which makes it more difficult for them to associate a cue and a reward if that cue has been unrewarded before. For example, bees will take longer to 'learn' that a certain floral odor signals a feeding opportunity if they first repeatedly encountered the smell when food was absent. Such a mechanism allows organisms to devote more attention to other stimuli which have the potential to be important for survival.

The strength of latent inhibition – as revealed by how quickly and easily an individual can learn to associate a reward with a previously unrewarded stimulus – can differ between individuals. For instance, this is the case in honey bee colonies, where workers have the same mother but may come from different fathers. Such genetic variation can be beneficial for the hive, with high latent inhibition workers being better suited for paying attention to and harvesting known resources, and their low latent inhibition peers for discovering new ones. However, the underlying genetic and neural mechanisms underpinning latent inhibition variability between individuals remained unclear.

To investigate this question, Latshaw et al. cross-bred bees from high and low latent inhibition genetic lines. The resulting progeny underwent behavioral tests, and the genome of low and high latent inhibition individuals was screened. These analyses revealed a candidate gene, *Amtyr1*, which was associated with individual variations in the learning mechanism.

Further experiments showed that blocking or disrupting the production the AMTYR1 protein led to altered latent inhibition behavior as well as dampened attention-related processing in recordings from the central nervous system. Based on these findings, a model was proposed detailing how varying degrees of *Amtyr1* activation can tune Hebbian plasticity, the brain mechanism that allows organisms to regulate associations between cues and events. Importantly, because of the way AMTYR1 acts in the nervous system, this modulatory role could go beyond latent inhibition, with the associated gene controlling the activity of a range of foraging-related behaviors. Genetic work in model organisms such as fruit flies would allow a more in-depth understanding of such network modulation.

capabilities (*Heyes, 2012*; *Dickinson, 2012*). This ability arises from various forms of associative and operant conditioning (*Mackintosh, 1983*). However, the absence of reward also provides important information for learning about stimuli, because all animals must use this information to redirect a limited attention capacity to more important stimuli (*Lubow, 1989*). One important mechanism for learning to ignore irrelevant stimuli is called latent inhibition (*Lubow, 1973*). After an animal is presented with a stimulus several times without reinforcement, learning is delayed or slower when that same stimulus is reinforced in a way that would normally produce robust excitatory conditioning. For example, when honey bees are repeatedly exposed to a floral odor without association to food rewards, their ability to subsequently learn an excitatory association of this odor with a reward is delayed or reduced (*Chandra et al., 2010*). While many studies in the honey bee have focused around how the presence of reward shapes learning and memory (*Langberg and Smith, 2006*), evaluating this important form of nonassociative learning has not received as much attention (*Chandra et al., 2010*; *Abramson and Bitterman, 1986*). Yet, like in all animals, it plays an important ecological role in the learning repertoire of honey bees. The presence of unrewarding flowers in an otherwise productive patch of flowers (*Seefeldt and De Marco, 2008*), or the unreinforced presence of an odor in the colony (*Fernández et al., 2009*), can influence foragers' choices of flowers during foraging trips.

Moreover, individual honey bees from the same colony differ in the degree to which they exhibit several learning traits (*Brandes, 1991*; *Chandra et al., 2000*; *Finke et al., 2021*; *Smith et al., 1991*; *Pamir et al., 2014*), including latent inhibition (*Chandra et al., 2000*). Several studies of different forms of learning have demonstrated that individual differences are heritable (*Chandra et al., 2000*). Individuals showing different learning phenotypes occur within the same colony because a queen mates with up to 20 drones (males) (*Page, 2013*), and thus honey bee colonies typically contain a mixture of many different paternal genotypes. This within-colony genetic diversity of learning capacities may

reflect a colony level trait that allows the colony to react and adapt to rapidly changing resource distributions (*Latshaw and Smith, 2005*; *Mosqueiro et al., 2017*).

Our objective here was to evaluate the genetic and neural mechanisms that underlie individual differences for latent inhibition in honey bees. We show that a major locus supporting individual differences maps to a location in the honey bee genome previously identified in independent mapping studies as being important for latent inhibition (*Chandra et al., 2001*) as well as for sugar and pollen preferences in foragers (*Hunt et al., 2007*; *Page et al., 2000*). Disruption of a tyramine receptor encoded by *Amtyr1* in this region changes expression of latent inhibition in a way that suggests that intact signaling via the *Amtyr1* pathway is important for modulating plasticity at inhibitory synapses. Furthermore, electrophysiological analyses combined with blockade of the AmTYR1 receptor in the antennal lobe – the first synaptic center along the olfactory pathway – decreased antennal lobe responsiveness to odor and blocked a neural correlate of latent inhibition. Finally, sequencing the gene failed to reveal mutations in the coding regions that would affect protein function, leading to the conclusion that variation across workers could arise from differential gene expression through transcriptional regulation.

We discuss how these data strongly imply a functional role for *Amtyr1* signaling in modulating expression of attention via latent inhibition. We use the term 'modulating' to specifically propose that *Amtyr1* is not causing latent inhibition. Rather, it modulates excitatory inputs to circuitry that implements hebbian plasticity between downstream components that drive latent inhibition. Specifically, disruption of *Amtyr1 increases* excitatory drive to those components, and that increase drives stronger inhibitory hebbian plasticity. We propose modifications to an existing model for LTP-based latent inhibition in the antennal lobe to show that it can produce both the high and low phenotypes in natural populations by simply increasing or decreasing the level of *Amtyr1* activation. This model also suggests how this gene can exert broad pleiotropic effects on several different behaviors by acting as a gain control in different types of neural circuits or physiological processes. Given the established

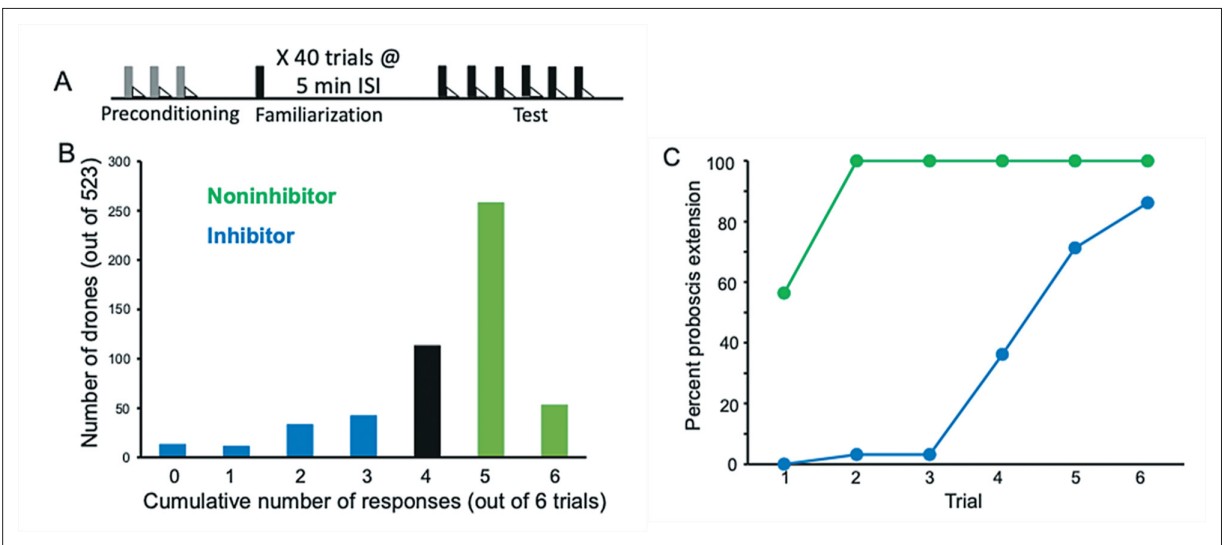

**Figure 1.** Evaluation and selection of drones from an F1 queen. (**A**) Drone honey bees were first evaluated by conditioning them over three trials to an odor (A; gray bars) followed by sucrose reinforcement (triangles) in a way that produces robust associative conditioning expressed as odor-induced proboscis extension response (PER) (*Bitterman et al., 1983*). All drones that showed no PER response on the first trial and PER response on each of the following two trials were selected for the subsequent familiarization phase. This procedure ensured that only drones motivated to respond to sucrose and learn the association with odor were selected. Approximately 10% of honey bees fail to show evidence of learning in PER conditioning using the collection methods described in Materials and methods. The familiarization phase involved 40 4 s exposures to a different odor (X; black bars) using a 5-min interstimulus interval. These conditions are sufficient for generating latent inhibition that lasts for at least 24 hr (*Chandra et al., 2010*). Finally, the test phase involved six exposures to X followed by sucrose reinforcement. (**B**) Frequency distribution of 523 drones evaluated in the test phase. The x-axis shows the summed number of responses over six conditioning trials. Fewer responses correspond to stronger latent inhibition. A total of 94 drones were selected in each tail of the distribution. 'Inhibitor' drones showed zero through three responses, and 'Noninhibitor' drones showed five or six responses. (**C**) Acquisition curves for the 94 inhibitor and noninhibitor drones. Approximately half of the noninhibitor drones showed spontaneous responses on the first trial, which is typical for noninhibitors in latent inhibition studies of honey bees (*Chandra et al., 2010*). All of the drones in that category showed responses on trials 1–6. In contrast, inhibitor drones showed delayed acquisition to the now familiar odor.

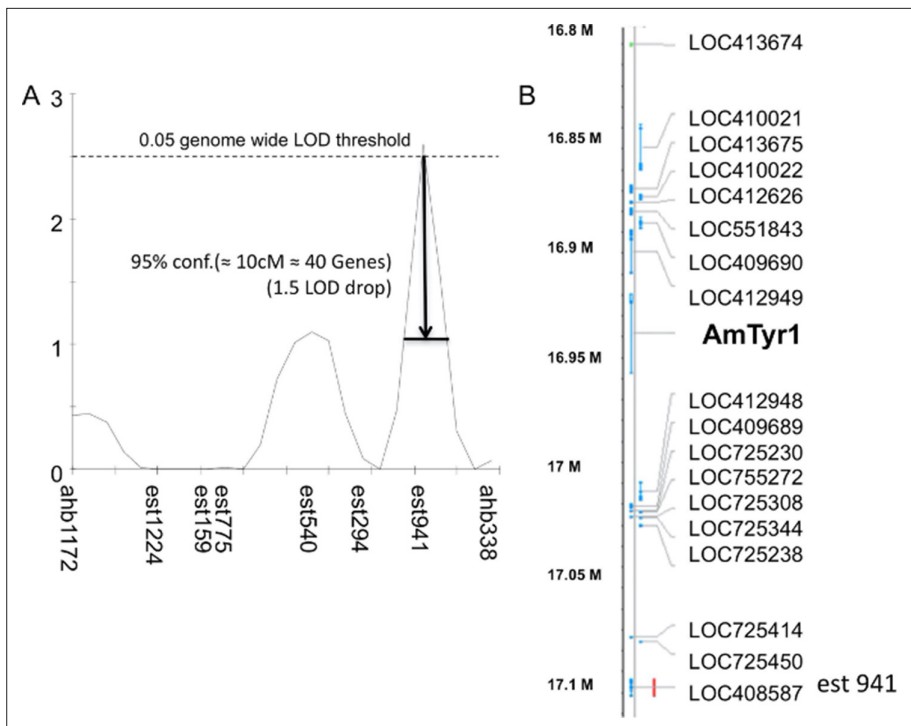

**Figure 2.** Single-nucleotide polymorphism (SNP) mapping of high and low recombinant drones. (**A**) Markers from linkage group 1.55 surrounding one significant Quantitative Trait Locus (QTL; est941 with a LOD score of 2.6). (**B**) Partial list of genes within 10 cM of the marker showing the location of *Amtyr1*.

*Amtyr1*-based variation between workers in how this behavior is expressed, and presumably in how the circuitry functions differentially in their brains, these findings are also important for understanding the strategies colonies use to explore for and exploit pollen and nectar resources (*Cook et al., 2020*).

## Results

We used two genetic lines of honey bees that had been bred for high (inhibitor) or low (noninhibitor) expression of latent inhibition. These lines were independently selected using identical methods to a previous study that had successfully bred high and low lines (*Chandra et al., 2001*). We evaluated 523 recombinant drones generated from a single hybrid queen produced from a cross between a drone from a noninhibitor line and a queen from an inhibitor line (*Figure 1*). Honey bee drones are ideal for behavior genetic studies because they are haploid progeny that develop from an unfertilized egg laid by the queen. We then selected 94 high and 94 low performing drones for the Quantitative Trait Locus (QTL) analysis, which identified one significant locus (*Figure 2A*). The QTL mapped to the same genomic region identified in a previous study of latent inhibition (called 'lrn1') using an independent inhibitor and noninhibitor cross and different (RAPD-based) genetic markers (*Chandra et al., 2001*). This is the same genomic region that has been identified in studies of foraging preferences of honey bees (*Page et al., 2000*; *Hunt et al., 1995*), where it has been called *pln2* for its effect on pollen versus nectar preferences and in modulating sensitivity to sucrose (*Pankiw et al., 2001*). Clearly, this genomic region has major effects on several foraging-related behaviors.

When we analyzed the gene list within the confidence intervals of this QTL, one gene – *Amtyr1* – in particular stood out (*Figure 2A, B*). That gene encodes a biogenic amine receptor for tyramine (AmTYR1) (*Blenau et al., 2000*) that is expressed in several regions of the honey bee brain (*Mustard et al., 2005*; *Sinakevitch et al., 2017*; *Thamm et al., 2017*). AmTYR1 is most closely related to the insect α2-adrenergic-like octopamine receptors and the vertebrate α2-adrenergic receptors (*Blenau et al., 2020*). Activation of AmTYR1 reduces cAMP levels in neurons that express it. We specifically considered AmTYR1 for more detailed evaluation for several reasons. Tyramine affects sucrose sensitivity in honey bees (*Scheiner et al., 2017*), and nurses and foragers differ in AmTYR1 expression

(*Scheiner et al., 2014*). Mutations in the orthologous tyramine receptor in fruit flies disrupt odor-guided innate behaviors to repellants (*Kutsukake et al., 2000*). Tyramine is also the direct biosynthetic precursor to octopamine (*Roeder, 2005*), which has been widely implicated in sucrose-driven appetitive reinforcement learning in the honey bee (*Farooqui et al., 2003*; *Hammer, 1993*). Therefore, Ventral Unpaired Medial neurons, which lie on the median of the subesophageal ganglion in the honey bee brain (*Sinakevitch et al., 2017*; *Sinakevitch et al., 2005*; *Sinakevitch et al., 2018*; *Kreissl et al., 1994*), and which form the basis for the appetitive reinforcement pathway must produce tyramine in the process of making octopamine. Recent analyses indicate these neurons in locusts and fruit flies also release both neuromodulators when activated (*Kononenko et al., 2009*; *Schützler et al., 2019*). Finally, octopamine and tyramine affect locomotor activity in the honey bee (*Fussnecker et al., 2006*).

We then evaluated whether nonsynonymous mutations in the coding sequence might change the functionality of the receptor. We performed a detailed genomic analysis of the 40-kb region including the *Amtyr1* gene, a 2-kb upstream, and a 0.5-kb downstream noncoding region. Single-nucleotide polymorphism (SNP) frequency in the coding sequence (CDS) was relatively low compared to the genome wide SNP frequency, and all 46 SNPs in the coding regions in any of the sequenced eight individual worker genomes represented synonymous substitutions, that is, these SNPs do not change the sequence of the encoded protein. Thus, phenotypic differences are not caused by structural changes in the tyramine receptor protein itself. We did, however, find an increased SNP frequency in introns, the up- and downstream noncoding regions and the 3' untranslated region. If *Amtyr1* is involved in latent inhibition, these variations might be linked to the changes in the regulation of *Amtyr1* gene expression, for example, by changes in transcription factor-binding sites or the stability of the mRNA, which might eventually be responsible for the observed phenotypic differences.

## Disruption of *Amtyr1* affects expression of latent inhibition

To further examine the role of *Amtyr1* signaling in latent inhibition, we performed a series of behavioral experiments that involved treatment of honey bees either with the tyramine receptor antagonist yohimbine (*Reim et al., 2017*) or with a Dicer-substrate small interfering (Dsi) RNA of the receptor (NCBI Reference Sequence: NM_001011594.1) to disrupt translation of mRNA into AmTYR1 (*Sinakevitch et al., 2017*; *Guo et al., 2018*). For these experiments, we used unselected worker honey bees from the same background population used for selection studies, which ensured that workers used for behavioral assays would represent a mixture of inhibitor and noninhibitor phenotypes. Therefore, treatment could increase or decrease the mean level of latent inhibition in this population. Training involved two phases (*Figure 3A*). First, during the 'familiarization' phase honey bees were identically exposed over 40 trials to odor X without reinforcement. Our previous studies have shown that this procedure produces robust latent inhibition. The second 'test' phase involved measurement of latent inhibition. During this phase odor X and a 'novel' odor N were presented on separate trials. Both odors were associated with sucrose reinforcement in a way that produces robust appetitive conditioning (*Bitterman et al., 1983*). Latent inhibition would be evident if responses to odor X were lower than the responses to the novel odor N. Injections of yohimbine directly into brains occurred either prior to the familiarization phase (*Figure 3A, B*) or prior to the test phase (*Figure 3C*).

The first experiment provided an important control procedure to evaluate whether yohimbine affects excitatory conditioning. This procedure involved familiarization to air, which does not induce latent inhibition to odor (*Chandra et al., 2010*). Honey bees familiarized to air learned the association of both odors with sucrose reinforcement equally well (*Figure 3A*). The response to each odor significantly increased, as expected, across trials ($X^2 = 47.5$, df = 3, p < 0.001). Moreover, there was no effect of injection with saline versus yohimbine; the response levels to all four odors across the saline and yohimbine injection groups were equivalent. Therefore, blockade of tyramine signaling does not affect excitatory conditioning, which is an important control for the effects about to be described. This control procedure also shows that yohimbine at $10^{-4}$ M probably does not affect receptors for other biogenic amines, such as octopamine, dopamine, and serotonin, all of which have been shown to have specific effects on appetitive olfactory learning in honey bees (*Farooqui et al., 2003*; *Wright et al., 2010*; *Hammer and Menzel, 1998*; *Mercer and Menzel, 1982*; *Bicker and Menzel, 1989*).

Yohimbine treatment affected the expression of latent inhibition in both treatments that involved familiarization to odor (the interaction between novel vs familiar odor and saline vs yohimbine injection:

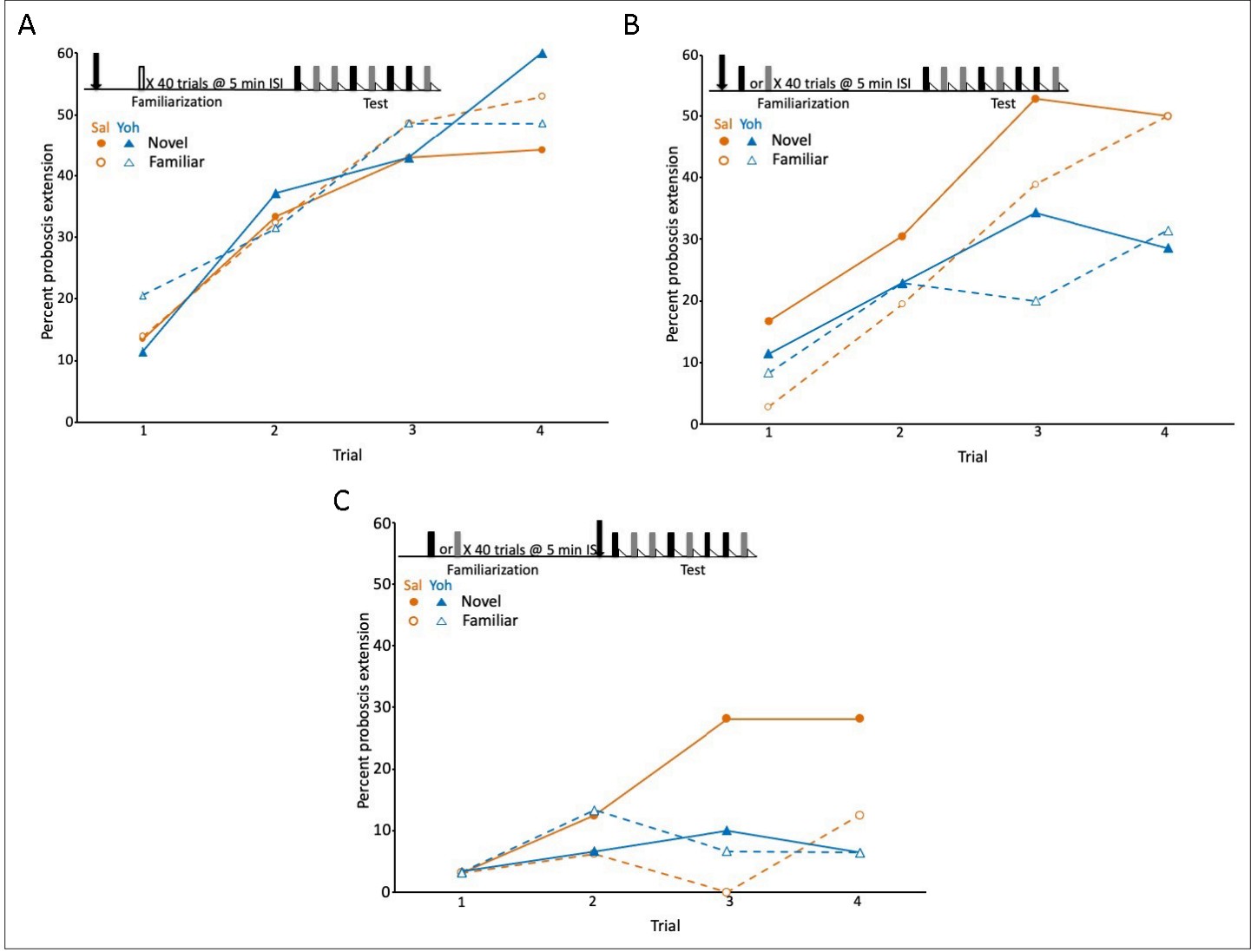

**Figure 3.** Blockade of the tyramine receptor with yohimbine modulated expression of latent inhibition. (**A**) Acquisition during the test phase in two injection groups of honey bees familiarized to air as a control procedure to evaluate the effects of yohimbine on excitatory conditioning. The conditioning protocol is shown at the top. In this experiment (and in **B** and **C**) we omitted the first phase (*Figure 1A*), which does not affect expression of latent inhibition (*Chandra et al., 2010*) and is only necessary when subjects are being selected for development of genetic lines. One group was injected (arrow) with saline (orange circles; *n* = 37 animals) and the other with yohimbine (blue triangles; *n* = 35) prior to familiarization. Because there was no odor presented during familiarization (open box), odors during the test phase were both 'novel' when conditioned, although one was arbitrarily assigned as familiar. The test phase in this experiment (also in **B** and **C**) differed from the test phase in *Figure 1*. For this design, each subject was equivalently conditioned to both odors on separate, pseudorandomly interspersed trials. Acquisition to both odors in both injection groups was evident as a significant effect of trial ($X^2$ = 47.5, df = 3, p < 0.001). None of the remaining effects (odor, injection, or any of the interaction terms) were significant (p > 0.05). (**B**) As in **A**, except both groups (orange saline: *n* = 36; blue yohimbine: *n* = 36) in this experiment were familiarized to odor; each odor (gray and black boxes; see Methods) was familiarized in approximately half of the animals in each injection group. In this design, each individual was equivalently conditioned to both odors during the test phase; latent inhibition is evident when the response to the novel odor is greater than to the familiar odor. Injection was prior to odor familiarization. (**C**) As in **B**, except injection of saline (*n* = 32) or yohimbine (*n* = 30) occurred prior to the test phase. Statistical analysis of datasets in **B** and **C** yielded a significant interaction ($X^2$ = 7.4, df = 1, p < 0.01) between injection (saline vs yohimbine) and odor (novel vs familiar) that was the same in both experiments, as judged by the lack of a significant odor × injection × experiment interaction term (p > 0.05). There was a higher response to the novel odor than to the familiar odor, but only in the saline injected groups. The lower rate of acquisition in **C** ($X^2$ = 64.0, 1, p < 0.01) could be due to performance of this experiment at a different time of year, or to injections immediately prior to testing, which affects levels proboscis extension response (PER) conditioning in honey bees but leaves intact relative differences between groups (*Gerber et al., 1996*).

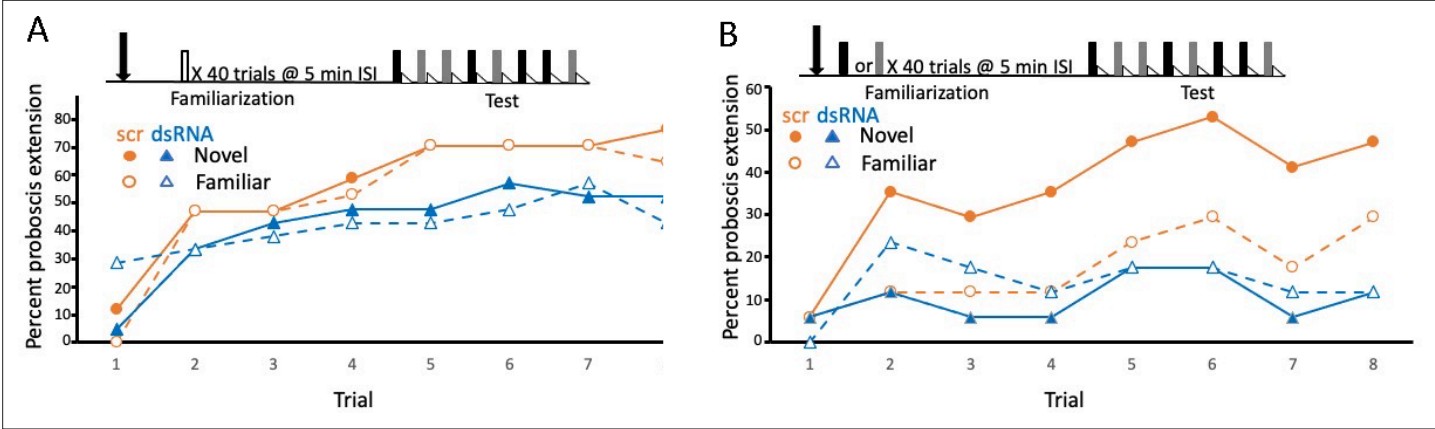

**Figure 4.** Disruption of translation of the tyramine receptor by DsiRNA also modulated expression of latent inhibition. This experiment was identical to that shown in *Figure 3A, B*, except injections were performed with a mixture of Dsi *Amtyr1*RNA (dsRNA; arrow) 24 hr prior to behavioral training and testing. The control for this experiment was a scrambled sequence of the *Amtyr1* RNA, Dsiscr (scr). (**A**) After treatment with Dsiscr (N = 17) or DsiRNA (N = 19) and familiarization to air, acquisition to both odors was significant across trials ($X^2$ = 62.7, 7, p << 0.01). There was also a significant effect of injection ($X^2$ = 8.8, 1, p < 0.01). However, the odor × injection interaction was not significant. (**B**) Same as in A, except familiarization was to odor (Dsiscr (N = 17) DsiRNA (N = 13)). The injection × odor interaction was significant ($X^2$ = 7.8, 1, p < 0.01). Quantitative PCR analysis of *Amtyr1* mRNA levels in brains revealed lower levels of mRNA in Dsi*Amtyr1* injected animals (0.046 ± 0.006) than in Dsiscr injected animals (0.142 ± 0.028).

$X^2$ = 7.4, df = 1, p < 0.01). First, in the saline controls, honey bees responded more often to odor N than to X after injection of saline prior to familiarization or prior to testing (*Figure 3B, C*, circles). The response to the familiar odor was lower than the response to the novel odor on most trials, including spontaneous responses on the first trial. Injection of yohimbine eliminated the difference in response to the novel and familiar odors. Moreover, the responses to both odors after yohimbine treatment were significantly lower than, or at least equal to, the response to the familiar odor in the respective saline controls. This pattern could not arise from blockade of excitatory learning about N, because excitatory learning was unaffected in the air preexposure controls (*Figure 3A*). Instead, the yohimbine-induced pattern was specific to the treatments in which one odor was familiar.

This result implies that blockade of AmTYR1 modulates latent inhibition to a familiar odor and that the effect now generalizes to the novel odor. Finally, the relative effect of yohimbine treatment, that is reduction of proboscis extension response (PER) rate, is similar when it is injected either prior to familiarization (*Figure 3B*) or prior to testing (*Figure 3C*). This pattern, that is, the same effect prior to acquisition or testing, is similar to the action of octopamine blockade on excitatory conditioning (*Farooqui et al., 2003*).

Although the results with yohimbine were promising, we were concerned that yohimbine can have effects on other receptors, specifically on an α2-adrenergic-like octopamine receptor (*Blenau et al., 2020*) and on an excitatory tyramine receptor AmTYR2 (*Reim et al., 2017*). Therefore, we decided to disrupt *Amtyr1* expression via injection of *Amtyr1* DsiRNA in order to provide an independent method to test the role of AmTYR1 in producing latent inhibition (*Figure 4*). Yohimbine blocks the receptor, whereas dsiAmtyr1 disrupts production of the receptor protein. Similar outcomes with the two different methods would increase confidence in the result. For the behavioral experiments, we used the same procedure as above for yohimbine except that the mixture of three *Amtyr1* DsiRNA constructs was injected 20 hr prior to conditioning because of the time frame needed for the DsiRNA to target mRNA. Because of that time frame, and because injection of yohimbine prior to either phase produced equivalent results, we performed injections of *Amtyr1* DsiRNA only prior to familiarization. As a control we used a scrambled sequence of *Amtyr1* (DsiScr). Use of DsiScr controls for possible nonspecific effects arising from any aspect of the injection.

Injection of *Amtyr1* DsiRNA produced the same effects as yohimbine. After familiarization to air as a control, both groups of foragers learned the association of both odors ($X^2$ = 62.7, 7, p << 0.01; *Figure 4A*), although there was a slight decrement in response rate in DsiRNA injected animals ($X^2$ = 62.7, 7, p << 0.01; see discussion below). In contrast, after familiarization to one of the odors, learning of both the novel and familiar odors was poor in the *Amtyr1* DsiRNA injected group (*Figure 4B*).

But expression of latent inhibition was normal – that is responses to the novel odor exceeded the responses to the familiar odor in the DsiScr group. As before the interaction between odor and injection was significant ($X^2$ = 7.8, 1, p < 0.01).

In conclusion, both behavioral experiments support the hypothesis that *Amtyr1* affects expression of latent inhibition without affecting excitatory conditioning. The results are dependent on unreinforced odor presentation, because that was the only difference between *Figure 3A–C* and between *Figure 4A, B*. However, the results at first glance seemed counterintuitive. Blockade and disruption of *Amtyr1* did not attenuate latent inhibition by, for example, increasing the responsiveness to the familiar odor. Instead, treatment with yohimbine or *Amtyr1* DsiRNA reduced responsiveness to the novel odor. This result is consistent with *Amtyr1* modulating inhibition involved in, for example, identified inhibitory processes in the antennal lobes and/or the mushroom bodies (*Linster et al., 2005*). Specifically, and as we propose below, it would prevent the inhibition from becoming too strong, and possibly keep it at a set point between very strong and very weak.

## Disruption of *Amtyr1* signaling affects neural codes for odors in the antennal lobe

Because of this intriguing result, we performed additional experiments to investigate the mechanism in more detail. Our prior studies of odor coding identified neural manifestations of latent inhibition in early synaptic processing of the antennal lobes of the honey bee brain (*Lei et al., 2022*; *Locatelli et al., 2013*). Familiarization to an odor X caused a mixture of a novel odor N and X to become much more like N (*Locatelli et al., 2013*). That is, neural information about familiar odors like X is filtered out of mixtures. Furthermore, responses to any novel odor are enhanced after familiarization to X (*Lei et al., 2022*), which is a form of novelty detection. These effects in the antennal lobes could arise because of expression of AmTYR1 in presynaptic terminals of sensory axons in the honey bee antennal lobes (*Sinakevitch et al., 2017*), where activation of AmTYR1 would decrease cAMP levels (*Blenau et al., 2000*) and likely decrease release of acetylcholine at synapses.

We therefore chose to analyze the effect of yohimbine treatment on odor processing in the antennal lobes by recording electrophysiological responses to odors prior to and after familiarization in combination with yohimbine treatment. We used yohimbine in these experiments because of the more rapid onset (minutes vs hours) compared to DsiRNA treatment. This first experiment did not employ familiarization to odor. Prior to yohimbine treatment, recordings from 71 units across 4 animals revealed responses to odors that ranged from no detectable change in spike activity with odor presentation to a robust increase in spiking activity (*Figure 5A*). After yohimbine treatment, responses decreased, although spiking activity was still detectable (*Figure 5B*). This decrease in response is consistent with AmTYR1 being involved in regulation of inhibition in networks of the antennal lobe, assuming most recorded units that showed a decrease were Projection Neurons (PNs). In our previous use of this technique approximately 45% of recorded units were PNs (*Lei et al., 2022*). Olfactory Receptor Neuron (ORN) spikes do not register on the electrodes. Hypothetically at least, when AmTYR1 is blocked by yohimbine, excitation of inhibitory networks in the antennal lobe increases and drives down PN responses.

We then evaluated whether continuous perfusion of the brain with saline or yohimbine *during odor familiarization* would interrupt how latent inhibition is manifested in the antennal lobe by potentiation of responses to novel odors, as we have reported (*Lei et al., 2022*). Indeed, yohimbine treatment modified how neurons respond to novelty. Using the same familiarization protocol as in *Figures 3 and 4*, but under conditions of saline perfusion, we found that 39% of units (N = 99) responded more strongly to the novel odor before the familiarization to an odor (*Figure 5C*, purple dots in upper panel; purple bar in *Figure 5D*). After familiarization, this percentage increased significantly to 54% (*Figure 5C*, orange dots in lower panel; orange bar in *Figure 5D*) (*McNemar* test with Yates's correction, df = 1, Chi-square = 5.939, p < 0.02). Hence, familiarization increased bias toward the novel odor in neurons that were more responsive to that odor to begin with, which is consistent with our earlier results (*Lei et al., 2022*). In different experiments where yohimbine was perfused, the familiarization protocol not only did not increase bias toward the novel odor, it significantly decreased the original bias from 49% (N = 56) to 14% (*Figure 5D*, gray bars) (*McNemar* test with Yates's correction, df = 1, Chi-square = 11.13, p < 0.001), suggesting that yohimbine interrupted this neural manifestation of latent inhibition in the antennal lobes.

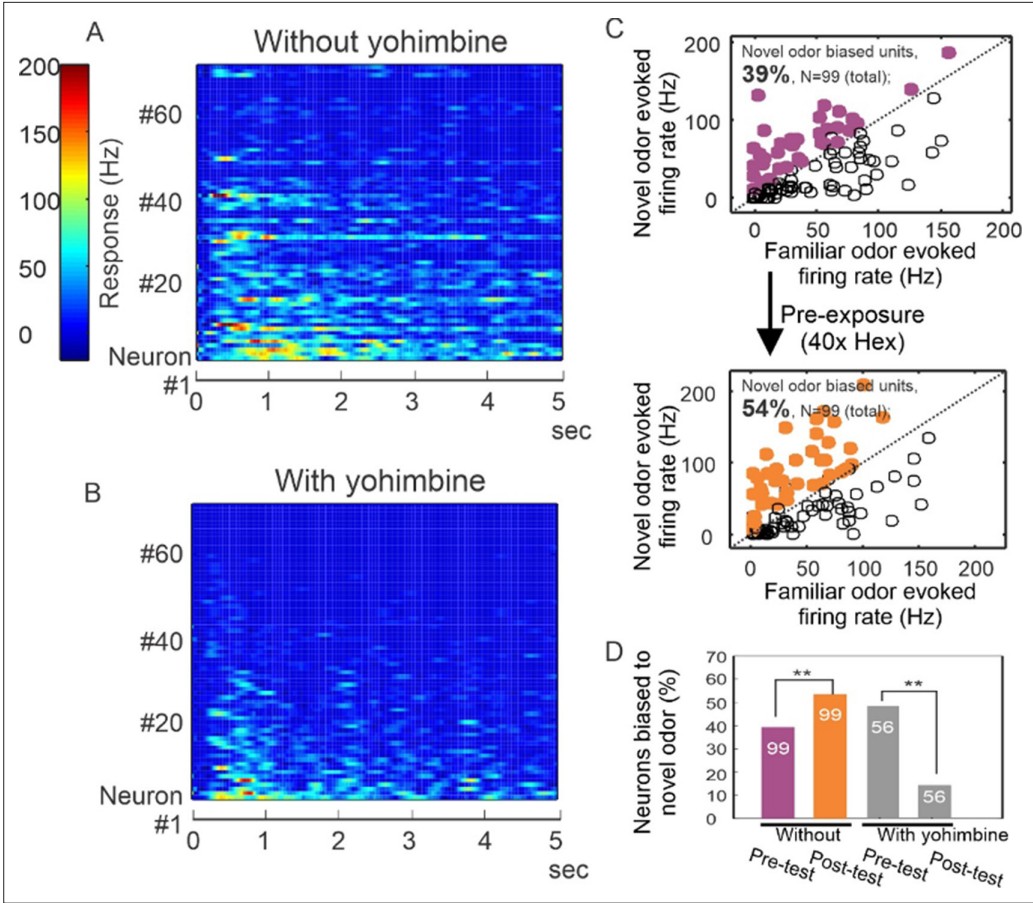

**Figure 5.** Yohimbine disrupts processing in the honey bee antennal lobe. (**A, B**) Perfusion of yohimbine solution (50 µM in physiological saline) into honey bee head capsule caused antennal lobe units to decrease response magnitude to odor stimuli (2-octanone and 1-hexanol) in general. Odor was delivered through a solenoid valve that was open at time zero and lasted for 4 s. (**C**) In control experiments where yohimbine was not applied, most of the units were responsive to both hexanol and octanone, but 39% were biased toward octanone (purple dots above the diagonal line), that is showing stronger response to octanone than to hexanol. During the familiarization protocol, these units were familiarized to hexanol 40 times with 1-min interval (arrow down) and were tested again with hexanol and octanone 10 min after the last odor stimulation in the familiarization phase. The test results show 54% of units responded more strongly to octanone (orange dots), which is a novel odor in this protocol. The 15% increase is statistically significant (*McNemar* test with Yates's correction, df = 1, Chi-square = 5.939, p < 0.02) (asterisks on purple and orange bars, N = 99). (**D**) When the familiarization protocol was used with saline versus yohimbine perfusion, the response bias toward novel odor was disrupted, showing a significant decrease in comparison with the familiar odor (*McNemar* test with Yates's correction, df = 1, Chi-square = 11.13, p < 0.001) (asterisks on gray bars, N = 56).

## The tyramine/octopamine ratio in the brain is also associated with latent inhibition

A recent report implicated the release of dopamine in driving reward seeking behavior (*Huang et al., 2022*). In order to evaluate whether dopamine might be involved in latent inhibition, and whether change in release of octopamine and/or tyramine might contribute to our behavioral results, we reanalyzed previously published data (*Cook et al., 2019*) on levels of dopamine, serotonin, octopamine, and tyramine in individual brains of 81 foragers collected from an unselected genetic background used for selection of lines for expression of latent inhibition. The foragers were collected as 'scouts' or 'recruits'. Scouts were defined as the first bees to explore a new landscape into which their colony had been moved. Recruits were defined as foragers that were exploiting resources once they were found. All scouts and recruits were trained for latent inhibition in the laboratory, and then classified as

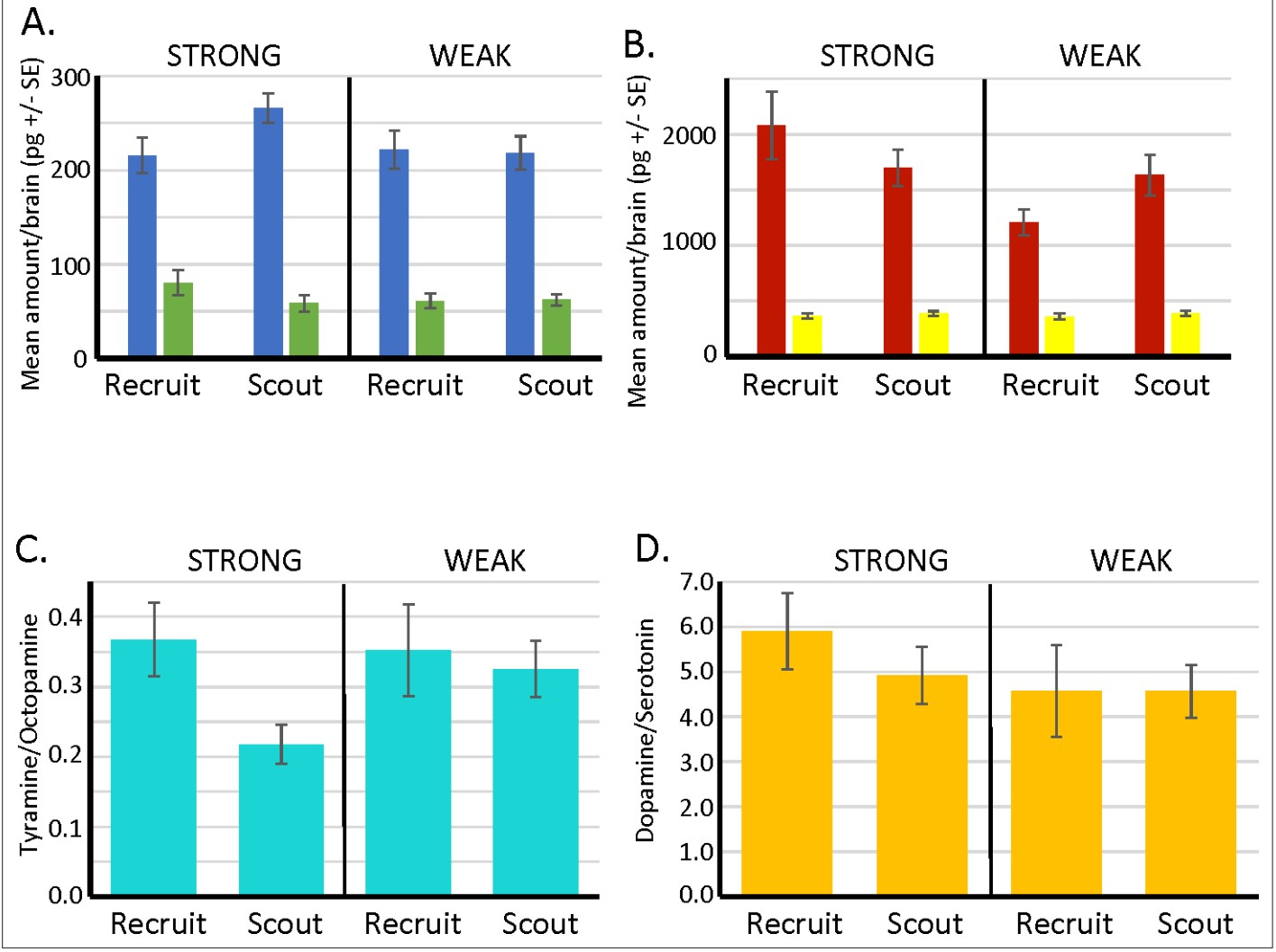

**Figure 6.** Biogenic amine levels in individual brains of scout and recruit foragers that expressed strong or weak latent inhibition. (**A, B**) Absolute levels of octopamine (blue) and tyramine (green) and of dopamine (red) and serotonin (yellow) in individual forager brains. (**C**) Ratios of tyramine/octopamine. In foragers that exhibited strong latent inhibition, the ratio was significantly lower in scouts ($N = 25$) than recruits ($N = 13$) (Wilcoxon $W = 56.0$, $p < 0.05$). Ratios did not differ in scouts ($N = 24$) and recruits ($N = 19$) that exhibited weak latent inhibition ($p > 0.05$). (**D**) Ratios of dopamine/serotonin did not differ in either the strong or weak groups ($p > 0.05$). Sample sizes the same as in C.

to whether they showed strong or weak latent inhibition based on learning a novel and familiar odor (*Cook et al., 2019*).

Of the biogenic amines (*Figure 6A, B*), only tyramine showed differences between scouts and recruits (see *Cook et al., 2019* for methods and a more complete analysis of these data). Differences in dopamine or serotonin levels were not significant. For the current purpose, we reanalyzed the data to focus on the ratios of tyramine to octopamine and dopamine to serotonin (serotonin was used a reference for dopamine levels in *Huang et al., 2022*). Scouts that showed strong latent inhibition also had significantly lower ratio of tyramine to octopamine than recruits, and that ratio was also lower than scouts and recruits that showed weak latent inhibition (*Figure 6C, D*). There were no significant differences in the dopamine to serotonin ratios. Thus, there is an interaction of tyramine and octopamine *production* with behavioral division of foraging labor and expression of latent inhibition. However, dopamine, serotonin, and their ratios do not appear to be involved in latent inhibition.

## Discussion

Our results have identified genetic and neural underpinnings that modulate an important form of learning and memory in the brain. All animals need to learn about stimuli in their environment. Latent inhibition is important for redirecting limited attention capacity away from unimportant, inconsequential stimuli and refocusing it toward novel stimuli about which the animal knows little or nothing. Two independent QTL mapping studies have now identified the genetic locus that contains *Amtyr1* as important for regulating individual variation in attention (*Chandra et al., 2001*). There are other loci in the genome that show associations with the behavior, and there are also other unidentified genes in the same locus. Nevertheless, our manipulation of *Amtyr1* function using both pharmacology and DsiRNA treatments confirm its association with behavioral expression of latent inhibition.

There are a few ideas that need to be kept in focus at this point in our understanding of *Amtyr1*. First, it is not a *latent inhibition gene*. Instead, it is a gene that has broad pleiotropic effects on foraging-related behaviors that include its effect on expression of latent inhibition. In that sense it has major effects on a broader behavioral syndrome that includes effects on sucrose sensitivity (*Pankiw et al., 2001*; *Thamm et al., 2017*; *Scheiner et al., 2017*), preferences for nectar and/or pollen (*Page et al., 2000*), behavioral caste differences (*Scheiner et al., 2014*), reproductive physiology (*Wang et al., 2020*), and learning (*Chandra et al., 2000*). The model we propose below, in which *Amtyr1* acts as a gain control on inputs to neural networks, could potentially explain how *Amtyr1* can have such broad effects.

Second, the effects of *Amtyr1* specifically on expression of latent inhibition likely arise by combining its expression in sensory as well as more central areas of the brain. That is, it is unlikely that there is a single locus in the brain that underlies latent inhibition. We have shown in honey bees (*Sinakevitch et al., 2017*), for example, that AmTYR1 is on presynaptic terminals of Olfactory Sensory Neuron axons in the antennal lobes, where they provide cholinergic excitation to dendrites of Local GABAergic Inhibitory Interneurons (LN) and PNs (*Figure 7A*). AmTYR1 is also on presynaptic terminals of PN axons that terminate in and also provide excitatory cholinergic inputs to the mushroom body calyces. In the antennal lobe, where our electrophysiological and imaging studies have focused (*Lei et al., 2022*; *Locatelli et al., 2013*; *Fernandez et al., 2009*; *Locatelli et al., 2016*), familiarization to an odor causes the neural representation of a mixture that contains the familiar odor to become more like novel odors in the mixture (*Locatelli et al., 2013*). Additionally, it potentiates the response to the novel odor (*Lei et al., 2022*). We assume, but have yet to show experimentally, that this bias combines with how AmTYR1 affects processing in the mushroom bodies, where olfactory information converges with information from other sensory modalities. These higher-order effects of AmTYR1 could underlie individual differences among genetic lines selected in the laboratory for odor-based latent inhibition when they show differential attention to sensory stimuli associated with feeders when tested in free flying conditions in the field (*Cook et al., 2020*).

The precise relationship of *Amtyr1* to latent inhibition is different from what is normally expected from disruption of a gene that underlies a behavior. We expected that disruption of *Amtyr1* function would reduce or eliminate latent inhibition; that is, learning about a familiar odor (X) would rise to equal learning about the novel odor. Instead, the response to the novel odor was reduced to equal that to the familiar odor. *This reduction was specific to familiarization treatment, so it is dependent on plasticity*. It cannot be explained by nonspecific – for example toxic – effects of treatment, because the same treatments did not reduce to the same extent excitatory conditioning in the absence of familiarization to an odor. Moreover, the same effect was evident using two very different means for disruption of *Amtyr1* signaling.

We propose that *Amtyr1* modulates neural plasticity in the antennal lobes and mushroom bodies that reduces attention to a familiar odor. *Amtyr1* maintains coactivation of LNs and PNs in the antennal lobe at a set point between the extremes where it becomes too strong (e.g. when *Amtyr1* is disrupted) or too weak (*Amtyr1* strongly activated). Given that activation of *Amtyr1* reduces cAMP levels, it would be expected that its activation would reduce excitability of axon terminals. Hypothetically then, activation of *Amtyr1* could reduce excitatory drive of post-synaptic processes on PNs and LNs in, for example, the antennal lobes, and possibly between PN axons and intrinsic and GABAergic extrinsic neurons of the mushroom bodies (*Sinakevitch et al., 2017*).

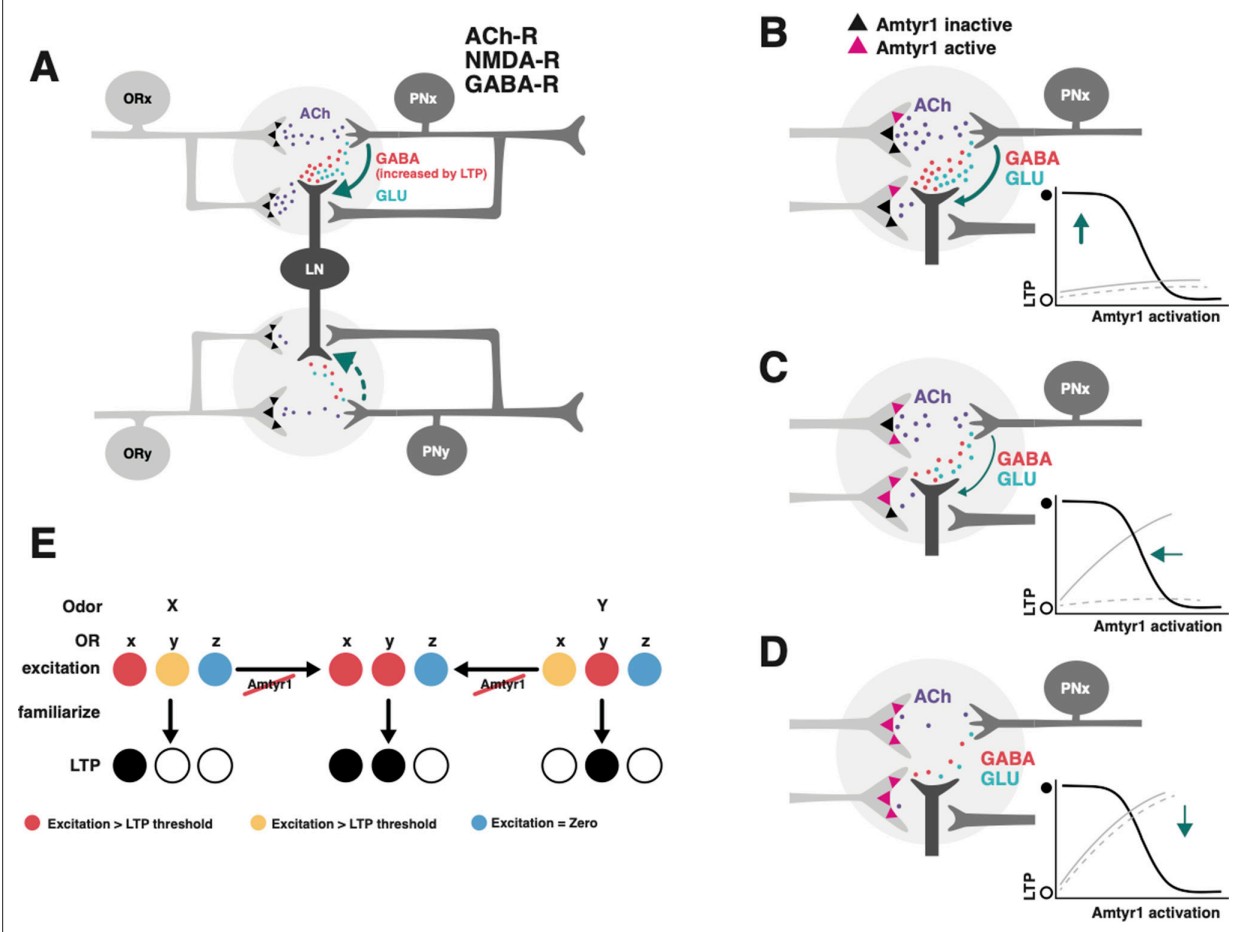

**Figure 7.** Hypothetical model of how *Amtyr1* could modulate hebbian plasticity inhibition to modulate latent inhibition. (**A**) Circuitry of two glomeruli (**A** has been adapted from Figure P1 from ***Das et al., 2011***) that underlies latent inhibition (habituation) in the fruit fly antennal lobe, and which was also proposed via a computational model to underlie changes in antennal lobe responses to familiar and novel odors in the honey bee (***Locatelli et al., 2013***). The only addition here is incorporation of *Amtyr1* receptors on Olfactory Receptor Neuron axon terminals (***Sinakevitch et al., 2017***). In the fruit fly (***Das et al., 2011***), ORNs release Ach (purple) to coactivate Projection Neurons (PNs) and Local GABAergic Inhibitory Interneurons (LNs), which both express ACh receptors. LNs release both GABA (red) and glutamate (green) onto synapses with PNs. NMDA receptors on PNs potentiate the LN/PN synapses via a retrograde signal (green arrows) to LNs to hypothetically increase the release of GABA. This hebbian plasticity increases inhibition of PNs by LNs and thus decreases excitation of the PN. LN/PN coactivation in other glomeruli is either nonexistent (e.g. ***Das et al., 2011***), too low to produce potentiation of the LN/PN synapse, or strong enough to produce inhibition to odors other than the familiar odor (dashed green arrow). AmTYR1 receptors on presynaptic terminals of ORNs (***Sinakevitch et al., 2017***) are shown as black triangles. Here, we do not show the source for tyramine. We assume it could be from the VUM neuron, which has terminals in the cortex of every glomerulus, where ORN terminals overlap with PN and LN dendrites (***Sinakevitch et al., 2013***). (**B–D**) Increasing activation of AmTYR1 (pink triangles) progressively decreases release of ACh and lowers coactivation of LNs and PNs. Decreased coactivation reaches a threshold (**D**) below which it fails to modify the LN/PN synapse, although ACh release might still activate the glomeruli. Graphs inset in each figure show the hypothetical relationship between activation of AmTYR1 (*x*-axis) verses LTP-based hebbian plasticity (*y*-axis; open and filled circles indicate low and high LTP, respectively). Low activation of AmTYR1 produces high coactivation-based LTP (**B**) and vice versa (**D**). Arrows show the hypothetical point on the LTP curve that represents LTP in each figure. Gray lines represent the hypothetical acquisition levels to novel (solid) and familiar (dashed) odors given the strength of hebbian plasticity in each figure. For these lines, the *x*-axis would be 'Trial' and *y*-axis 'Percent proboscis extension', as in ***Figures 3 and 4***. (**E**) Hypothetical relationship between overlapping odor coding, *Amtyr1* activity and LTP across three ORN types in the antennal lobe. ORNs x and y show coding with different levels of activity (red, yellow, blue high to no activity) for odors X and Y. Disruption of *Amtyr1* increases activity in x and y but not in a third ORN(z), which shows no activity to either odor. LTP is generated whenever activity reaches a 'red' threshold.

## A model of how *Amtyr1* could modulate inhibition in antennal lobe networks

We can now propose a model for how *Amtyr1* could act in the antennal lobes, and possibly also as an important component for regulating latent inhibition in distributed networks in the mushroom body.

The model would have to explain why air (mechanosensory) stimulation alone does not reduce subsequent acquisition to odors (*Figures 3A and 4A*) as much as familiarization to an odor (*Figures 3B, C and 4B*). Mechanosensory stimulation produces fast, transient responses in antennal lobe glomeruli, whereas odor stimulation produces more robust, longer lasting responses (*Tuckman et al., 2021a*; *Tuckman et al., 2021b*). These differences in response could underlie the difference in response to air versus odor in our analyses, particularly in driving activity dependent plasticity at LN-to-PN synapses that we describe below.

We propose that the *Amtyr1*-based effects *specific to odor familiarization* occur by amplifying hebbian plasticity between PNs and LNs in the antennal lobe networks (*Figure 7A*). A computational model of the honey bee antennal lobe previously identified hebbian plasticity at Local GABAergic Inhibitory Interneuron (LN) synapses onto PNs as the most likely locus of plasticity to give rise to the observed biasing of an odor mixture to be less like a familiar odor and more like a novel odor (*Locatelli et al., 2013*). If ORy in *Figure 7A* is not activated by an odor (X) that activates ORx, then, after familiarization to X, a novel odor that activates ORy would be better able to suppress X (via lateral inhibition) in a mixture because of strengthened LN-to-PNx synapses (see schematic in Figure 7 of *Locatelli et al., 2013*). A potentiated novel odor would strengthen this effect. Work with fruit flies also showed that hebbian plasticity at the same synapses underlies odor habituation (*Das et al., 2011*; the same process as latent inhibition just differently named).

We now use the same model framework (see schematic in Figure 7 of *Locatelli et al., 2013*) to propose how Amtyr1 could affect neural networks in the antennal lobe and mushroom bodies. In our new model (*Figure 7A–E*), as in *Locatelli et al., 2013*, coactivation of PNs and LNs via excitation from OR axon terminals would produce plasticity at the LN-to-PN synapses. Low activation of AmTYR1 (*Figure 7A, B*) would lead to strong input from sensory axon terminals that would maximally activate LNs and PNs, thus leading to strong hebbian plasticity at LN-to-PN synapses. That is, it would lead to the strong inhibition of PNs in our manipulations that blocked or disrupted AmTYR1 signaling. Our model represents new now testable hypotheses. At moderate levels (*Figure 7C*) we propose that this plasticity could give rise to normal latent inhibition in our high attention genetic lines just as it does in fruit flies. Strong activation of AmTYR1 would weaken the activity and reduce or prevent plasticity (*Figure 7D*), and hence lead to learning about both the novel and familiar odors (such as in our low attention genetic lines that do not show strong latent inhibition; *Chandra et al., 2000*). In this latter case, activation of the antennal lobe and mushroom bodies by odors might be too low to induce LTP (*Figure 7E*; all loci would be below red) but still high enough to support odor detection, discrimination, and learning.

Furthermore, combinatorial coding of odors with disruption of AmTYR1 signaling might cause generalization of latent inhibition to novel odors, as we have observed. Many of the monomolecular odorants that have been used to study olfaction and olfactory learning in honey bees have neural activity patterns that partially overlap (*Paoli and Galizia, 2021*), such as with the representations for hexanol and 2-octanone. For each odor, PNs in a few of the 160+ glomeruli of the antennal lobe are highly activated, and a subset of other PNs are activated to a lesser degree. Hypothetically at least, if the familiar odor activates some of the ORs from glomeruli that also code for the novel odor, which is likely, then coactivation of those PNs with lateral inhibition could cause hebbian plasticity at those synapses too. Under normal circumstances the excitation of those ORs might be too low to potentiate inhibition (*Figure 7C–E*). But when AmTYR1 signaling is too low or disrupted (*Figure 7A, B*), ORN-driven coactivation of PNs and LNs would increase enough to drive the plasticity and reduce responses to novel odors (*Figures 3B, C, 4B, and 7E*). In this case, the discriminability of familiar and novel odors could be reduced or eliminated by the Hebbian plasticity.

We have linked expression of latent inhibition to hebbian plasticity in synapses from inhibitory LNs to PNs in the antennal lobe. Although any behavioral phenomenon likely arises from distributed neural networks in the brain, we focused on the effect of familiarization in the antennal lobes because of our prior analyses of odor processing and latent inhibition there in the honey bee (*Locatelli et al., 2013*), and because of reports of latent inhibition in the same networks in the fruit fly (*Das et al., 2011*). We do not specifically identify the type of LN represented in *Figure 7*, but we speculate that it would belong to the group of heteroLNs (*Fonta et al., 1993*), which receive excitatory inputs in one glomerulus and broadly transmit inhibition across glomeruli. Computational modeling suggests that the interglomerular connectivity of heteroLNs should be based on 'functional networks' defined by

overlapping glomerular activity patterns to similar odors (*Linster et al., 2005*). However, *Figure 7* only represents a minimal part of the network that we feel is needed to convey a hypothetical modulation of hebbian plasticity. It will be useful to consider the broader network as represented in *Sinakevitch et al., 2017* to more clearly understand how AmTYR1 functions in the antennal lobe, and how it may function in the calyces of the mushroom bodies as well.

We show in *Figure 5* that treatment with yohimbine to block AmTYR1 reduced responses to odor, which is consistent with increased excitation from ORN axon terminals driving inhibition, as represented in *Figure 7*. In fact, this reduction (in the antennal lobe and possibly the mushroom bodies) could be the reason for slightly lower acquisition in *Figure 4A* under dsRNA treatment. If true, the reduction in unit responsiveness did not completely block excitatory conditioning to odors or the expected potentiation after latent inhibition treatment.

We show the effect of odor familiarization on activity recorded from the antennal lobe as a potentiation of the response to a novel odor, which is consistent with our earlier report (*Lei et al., 2022*). AmTYR1 block, when coupled to familiarization, decreased responses to the novel odor relative to the potentiation normally observed in controls. Although we show that block of potentiation after familiarization depends on activation state of AmTYR1, we do not, and at this point for lack of data we cannot, represent this mechanism in *Figure 7*. Potentiation could occur via an as yet unknown process intrinsic to the antennal lobe neural networks (*Sinakevitch et al., 2017*). Alternatively, it could occur via identified feedback pathways to the antennal lobe (*Kirschner et al., 2006*; *Hu et al., 2010*) from neural mechanisms in the mushroom bodies that are known to produce potentiation to novel stimuli in fruit flies (*Hattori et al., 2017*). Under normal circumstances AmTYR1 is functional and moderating the hebbian plasticity at LN-to-PN synapses at levels consistent with *Figure 7C, D*, where novel odors are learned well (likely aided by potentiation). Blocking or disrupting AmTYR1 puts the network in a state consistent with *Figure 7B*, where responses to both types of odors are affected by hebbian plasticity at LN-to-PN synapses – including potentiated novel odors given the overlap in sensory representations.

It remains to be determined what the source for tyramine could be, and which of the local interneuron types might be the ones mediating latent inhibition in the antennal lobes. VUM neurons are an obvious possibility for the source of tyramine, since tyramine is a direct precursor to octopamine released by VUM neurons (*Roeder, 2005*). However, this hypothesis would depend on tyramine being released at almost constant, low levels without stimulation of VUM by taste receptors sensitive to sugars.

Finally, we have presented a heuristic, verbal model designed to summarize what we know about where *Amtyr1* is in the brain, how *Amtyr1* works by reducing cAMP levels, and how it could interact with established mechanisms of Hebbian plasticity between PNs and LNs that underlie latent inhibition. We feel it can predict the two natural behavioral phenotypes we find within honey bee colonies

> ## Box 1. Important questions that need to be addressed in honey bees and other animal models, such as the fruit fly, and in computational models:
>
> - To what extent is tyramine constantly released at a background level, such that it modulates activity of AmTYR1?
> - What is the balance of octopamine and tyramine during odor stimulation and when the odor is associated with reinforcement?
> - How does the action of AmTYR1 in distributed networks, such as the antennal lobes and mushroom bodies, coordinate to produce behavioral expression of latent inhibition (*Figure 7*)?
> - How is *amtyr1* activity regulated by other genes in a network, and by epigenetic factors in the environment?
> - Can the gain control model for *amtyr1* be extended to account for pleiotropic effects on other behaviors?

as well as the experimental results of disruption of Amtyr1 function. But this prediction lies on testable assumptions. For example, we assume that the level of LTP that develops will increase with increases in excitation gated by *Amtyr1*. How well our model works might also depend on a nonlinear thresholding function for coactivation that drives Hebbian plasticity (AmTYR1–LTP relationships and expected learning curves shown in *Figure 7B–E*). These and other parameters will need to be investigated both experimentally and computationally to more fully evaluate how the model applies to antennal lobe and mushroom body function, and whether and under what conditions it will work.

## Ideas and speculation: gain control and modulator ratios as modes of action

The modulatory role that we propose for *Amtyr1* could help to explain its broad pleiotropic effects on many different behaviors. We propose that AmTYR1 acts as a kind of gain control to regulate activity in any neural network it is providing inputs to. Differing degrees of *Amtyr1* activation in different neural circuits in the central or peripheral nervous system might regulate activity in those circuits to drive behaviors in one direction or the other; for example, toward high or low sensitivities to sucrose (*Scheiner et al., 2017*), preferences for pollen versus nectar (*Hunt et al., 1995*), and states of worker reproductive physiology (*Wang et al., 2020*).

Our model for the antennal lobes and mushroom bodies is reminiscent of recent analogous findings involving gain control in select forms of mammalian learning (*Fu et al., 2014*; *Fu et al., 2015*). Cholinergic regulation of a disinhibitory circuit within the mouse visual cortex has been shown to regulate cortical gain control, plasticity, and learning. Understanding the dynamic mechanisms underlying network modulation across multiple model organisms may shed light on robust and similar circuit motifs for various behaviors.

We were initially drawn to *Amtyr1* because of its relationship potentially to the release of tyramine by identified VUM neurons, which have been implicated in excitatory conditioning through release of octopamine (*Farooqui et al., 2003*; *Hammer, 1993*). VUM neurons must make tyramine in the process of making octopamine, and they likely release both biogenic amines. In particular, the dynamic balance between octopamine and tyramine is important for regulating insect behaviors (*Kononenko et al., 2009*; *Schützler et al., 2019*). It is intriguing to now propose and eventually test whether a balance between octopamine and tyramine release from VUM neurons is critical for driving attention in one direction or another depending on association with reinforcing contexts. In this model, activation of VUM neurons would release octopamine to drive excitatory association between odor and reinforcement. At the same time, release of tyramine would suppress excitatory drive onto inhibition. Both processes could synergistically drive the association. Furthermore, if there is a low level of background tyramine release from VUM when unstimulated, it would explain why in the *Amtyr1* DsiRNA injected group in *Figure 4A* responded slightly lower than the Dsiscr control group.

Interestingly, we have identified a potential interaction in the ratio of tyramine to octopamine between foraging role (scouts vs recruits) and expression of latent inhibition. The lower tyramine-to-octopamine ratios in scouts would potentially activate this receptor even less that it would normally be, yielding stronger inhibition according to the model described above. Further analyses are needed to test this prediction in more detail and evaluate its role in the foraging ecology of honey bees.

Finally, why do individuals in colonies under quasi-natural conditions differ in expression of latent inhibition, and presumably in the functioning of *Amtyr1*? We have used this naturally occurring and selectable genetic variation to establish colonies composed of different mixtures of genotypes (*Cook et al., 2020*; *Smith and Cook, 2020*). The mixture of genotypes in the colony affects whether and how quickly colonies discover new resources via an attention-like process operant in individual foragers (*Smith and Cook, 2020*). We have therefore proposed that genetic variation leading to colony level variation in *Amtyr1* expression represents a balance between exploration for and exploitation of resources. The precise balance of genotypes would give colonies flexibility to respond to changing resource distributions over the life of the colony.

## Materials and methods

### Selection of honey bee lines for differences in latent inhibition

We established high and low latent inhibition lines by conditioning drone and virgin queen honey bees to odors in three different conditioning phases (*Chandra et al., 2000*). The first phase involved selection of drones or queens that could successfully learn to associate an odor with sucrose reinforcement, which established that the honey bees were motivated to learn. This initial excitatory conditioning does not affect generation or expression of latent inhibition. The second 'familiarization' phase involved 40 unreinforced odor exposures for 4 s each; this new odor (black box; X) was different and discriminable from the first odor. The third and final phase involved conditioning honey bees to the now familiar odor X associated with sucrose reinforcement in a way that normally produces robust associative conditioning (*Bitterman et al., 1983*). Strong latent inhibition should slow the rate of learning to X. Drones and queens that exhibited this 'inhibitor' phenotype (defined as zero or one response to X over six conditioning trials) (*Chandra et al., 2000*) were mated using standard instrumental insemination techniques (*Cobey et al., 2013*) for honey bees to create a high (inhibitor) latent inhibition line. Drones and queens that learned X quickly (five positive responses to X over six trials) were also mated to produce a low (noninhibitor) latent inhibition line. Our previous studies have shown that worker progeny from inhibitor and noninhibitor matings showed significant correlation in expression of latent inhibition to their parents (*Chandra et al., 2000*; *Ferguson et al., 2001*).

### Recombinant drones

Male honey bees (drones) were produced from a cross between genetic lines selected for high and low expression of latent inhibition (*Chandra et al., 2000*; *Latshaw and Smith, 2005*). Hybrid queens were reared from a cross of a queen from the inhibitor line instrumentally inseminated (*Cobey, 2007*) with sperm from a single drone from the noninhibitor line. These queens were then allowed to mate naturally to increase longevity in a colony. Natural mating involves mating with several different drones. However, since drones arise from unfertilized eggs, the haploid (drone) genotype involves only recombination of the genotypes of the high and low lines in the hybrid queen. A single hybrid queen was then selected to produce drones. Sealed drone brood from the hybrid queen was placed in a small nucleus colony. Queen excluder material (wire mesh that does not permit the passage of queens or drones) was used to confine the emerging drones to the upper story. Upon emergence, drones were individually marked on the thorax with enamel paint for later identification, and then marked drones were co-fostered in a single outdoor colony until collected for behavioral conditioning.

Mature drones were collected from the colony upon returning from mating flights during the late afternoon the day before testing. Returning drones gathered on a piece of queen excluder material blocking the colony entrance and were put into small wooden boxes with queen excluder material on each side. They were then fed a small amount of honey and placed in a queenless colony overnight. The following morning drones were secured in a plastic harness using a small piece of duct tape (2 mm × 20 mm) placed between the head and the thorax (*Bitterman et al., 1983*). All drones were then kept at room temperature for 2 hr. They were then screened for their motivation to feed by lightly touching a small drop of 2.0 M sucrose solution to the antennae. Drones that extended their proboscis were selected for training.

### Foragers

Female pollen foragers (workers) were captured at the colony entrance as described above. Each bee was chilled to 4°C, restrained in a harness and fed to satiation with 1.0 M sucrose. The next day bees were tested for motivation by stimulation of their antennae with 2.0 M sucrose; bees that extended their proboscis were used in experiments shown in *Figures 3 and 4*.

### Conditioning protocols

#### Familiarization

Familiarization to the odor was done as described in *Chandra et al., 2010*. Restrained bees were placed in individual stalls where a series of valves regulated odor delivery via a programmable logic controller (PLC) (Automation Direct). Hexanol and 2-octanone were used either as pure odorants or diluted to 2.0 M in hexanes with odor treatments counterbalanced across animals. Odor cartridges

were made by applying 3.0 µl of odorant onto a piece of filter paper (2.5 × 35 mm) and inserting the filter paper into a 1-ml glass syringe. The odor cartridge was then connected to a valve regulated by the PLC that shunted air through the cartridge for 4 s once the automated sequence was initiated. Odor preexposure in all experiments involved 40 unreinforced presentations of odor for 4 s using a 5 min (*Figure 1*) or 30 s (*Figure 3*) intertrial interval (ITI). All odor cartridges were changed for fresh ones after every 10 uses to avoid odor depletion (*Smith and Burden, 2014*). The use of pharmacological treatment necessitated the use of a shorter ITI to avoid having the drug wear off before the end of preexposure. Our previous studies have revealed that latent inhibition is robust over this range of ITIs and odor concentrations (*Chandra et al., 2010*).

### PER conditioning

All PER learning paradigms used for testing used a 5-min ITI. An acquisition trial consisted of a 4-s presentation of an odor, the conditioned stimulus (CS, black or gray bars), followed by presentation of a 0.4-µl drop of 1.0 M sucrose solution, the unconditioned stimulus (US, triangles in *Figures 1, 3, and 4*). Three seconds after onset of the CS the US was delivered using a Gilmont micrometer syringe. The US was initially delivered by gently touching the antennae to elicit proboscis extension and subsequent feeding. Once a bee began to extend its proboscis at the onset of CS delivery, it was no longer necessary to touch the antennae prior to feeding.

We used two different procedures for testing latent inhibition after familiarization. For evaluation of recombinant drones (*Figure 1*), subjects were conditioned to the familiarized odor (X) as the CS over 6 forward pairing trials. The second procedure (*Figures 3 and 4*) involved use of a within animal control protocol. After familiarization all subjects received equivalent PER conditioning to two odors, one was the familiarized odor (X) and the other was a novel odor (N) that honey bees can easily discriminate from the familiarized odor (*Smith and Menzel, 1989*). Odors were presented in a pseudorandomized order (NXXNXNNX or XNNXNXXN) across trials such that equal numbers of animals received N or X on the first trial. Pharmacological treatment required the use of a control procedure involving familiarization to air to evaluate the degree to which expression of excitatory conditioning was affected by drug treatment (*Figures 3A and 4A*).

## Linkage analysis

Upon completion of the training paradigm, 523 drones were placed in individual 1-ml micro-centrifuge tubes and stored at −70°C. Genomic DNA extraction followed a standard protocol developed for honey bees (*Hunt and Page, 1995*). For SNP analysis, DNA was selected from 94 drones that exhibited the highest level of latent inhibition (0, 1, 2, or 3 responses over the six test trials) and from another 94 drones that exhibited the lowest level (5 or 6 responses). Analysis of the 188 samples was conducted by Sequenom, Inc, San Diego, CA.

The linkage map was built with a set of 311 SNP markers. The list of selected markers was provided by Olav Rueppell from previous studies examining the genetic architecture of foraging behavior and sucrose response thresholds (*Rueppell et al., 2006*; *Rueppell et al., 2004*). The 74 SNPs segregating in our mapping population were used for a QTL analysis. Map positions for markers in linkage group one were determined using the *Apis mellifera* 4.0 genome. The software MultiPoint 1.2 (http://www.mulitqtl.com) was used to determine the actual recombination frequencies for markers in linkage group 1. Recombination frequencies were then converted to centiMorgans using the Kosambi mapping function. The actual mapping distances in our mapping population were used in the QTL analysis. QTL analysis was performed with MapQTL 4.0. Interval mapping and MQM mapping revealed one significant QTL. Genomewide significant thresholds for $p < 0.05$ (LOD = 2.6) and $p < 0.01$ (LOD = 3.2) were determined using an implemented permutation test (1000 runs).

## Pharmacological and DsiAmTyr1 treatments

Yohimbine hydrochloride (Sigma) was diluted to $10^{-4}$ M in saline (5 mM KCl, 10 mM $NaH_2PO_4$, pH 7.8). We chose a concentration of yohimbine that has been shown to be effective in our previous study of its effect on honey bee behavior (*Fussnecker et al., 2006*). One µl of drug or saline alone was injected into the brain through the median ocellus using a Hamilton syringe (Hamilton; Reno, NV). Training began 15 min after injection, as this time has been shown to be effective in other drug studies using the same methodology (*Chandra et al., 2010*; *Mercer and Menzel, 1982*; *Menzel et al., 1999*).

**Table 1.** Nucleotide sequences of sense and antisense strands of control DsiSCR and AmTyr1 DsiRNA.

| DsiRNA | Sequences |
| --- | --- |
| DsiScr | 5'-GAGUCCUAAGUUAACCAAGUCACAGCA-3' 3'-CUCAGGAUUCAAUUGGUUCAGUGUCGU-5' |
| DsiTyr1_N | 5'-AGCGUGACGUUGGAUUGACGAGAGC-3' 3'-CCUCGCACUGCAACCUAACUGCUCUCG-5' |
| DsiTyr1_T1 | 5'-CCUGUGCAAAUUGUGGCUAACCUGC-3' 3'-GUGGACACGUUUAACACCGAUUGGACG-5' |
| DsiTyr_C | 5'-CAACGCUUGUUUAUUGCAUCUAUCG-3' 3'-CCGUUGCGAACAAAUAACGUAGAUAGC-5' |

For DsiRNA studies, we used sequences and protocols developed previously for a study of *Amtyr1* receptor distribution in the brain, which in that study were used to show that the anti-*Amtyr1* antibodies specifically recognized the receptor (*Sinakevitch et al., 2017*). We used a Dsi RNA of the *AmTyr1* receptor (NCBI Reference Sequence: NM_001011594.1) to knockdown *AmTyr1* mRNA receptor in the brain. We used the mixture of three DsiAmTyr1 constructs designed by the tool in IDT technology (*Sinakevitch et al., 2017*; *Table 1*). As a control we used a scrambled (dsiScr) version of the *Amtyr1* sequence. A 138 nanoliter injection of a 100-µM mixture of dsiAmTyr1 or dsiScr (Nanoinject 2000) was made into the middle ocellus 18–20 hr before behavioral tests. All injections were done blind so that the investigator doing behavioral tests was not aware of the content of the injection. After the tests brains without optic lobes were dissected out and homogenized each in TRIzol (Invitrogen) ($N$ = 27 for bees injected with dsiScr and $N$ = 32 for bees injected with DsiAmTyr1). Then, the total mRNA from each injected brain was extracted separately using the manufacturer's protocol for TRIzol method (Invitrogen). Contaminating genomic DNA was removed using DNA-free kit (Ambion, AM1906). RNA quantity and purity were evaluated using a NanoDrop (NanoDrop 2000). Expression of AmTyr1 was quantified using QuantiFAST SYBR Green RT-PCR kit (QIAGEN) on Applied Biosystem 7900 cycler (ASU DNA Facilities) with the protocol provided by the kit for a 384-well plate. The primers for quantitative real-time PCR assays were: AmTyr1_F 5'- GTTCGTCGTATGCTGGTTGC-3', AmTyr1_R 5'- GTAGATGAGCGGGTTGAGGG-3' and for reference gene AmActin_F 5'- TGCCAACACTGTCCTTTCTG-3', AmActin_R 5'- AGAATTGACCCACCAATCCA-3' (*Tuzmen et al., 2007*).

All injections were done blind so that the investigator doing behavioral tests was not aware of the content of the injection.

## Electrophysiological recordings from the antennal lobe

Extracellular recordings were performed in the antennal lobes with a 16-channel probe (NeuroNexus, Ann Arbor, MI). Spike waveforms were digitized with a RZ2 system at a sampling rate of 20 kHz (Tucker-Davis Technologies, Alachua, FL). After a stable recording was achieved, the honey bee preparation was first stimulated with two presentations of each of the following odors: 1-hexanol (Hex) and 2-octanone (Oct). The duration of each pulse was 4 s, and 2 min of recovering time were allowed between two pulses. During the preexposure phase, 40 pulses of Oct were delivered with inter-pulse interval of 60 s, after which 10-min recovery was given before testing. Upon completion of each experiment, extracellular spike waveforms are exported to Offline Sorter program (Plexon Inc, Dallas, TX) which classifies the similar waveforms into individual clusters (units) based on the relatedness of waveforms' projection onto a 3D space derived from the first three principle components that capture the most variation of the original waveforms. To increase the discriminating power, the original waveforms are grouped in a tetrode configuration, matching the physical design of the recording probe, that is 16 recording sites are distributed in two shanks in a block design of 2 × 4. Each block is called a tetrode. Statistical separation of waveform clusters, representing individual neurons or units, is aided with visual inspection, all implemented in the Offline Sorter program. Once satisfied with the clustering results, the time stamps of waveforms are then exported to Neuroexplorer program (Plexon Inc, Dallas, TX) and Matlab (Mathworks, Natick, MA) for further analysis.

Yohimbine (Millipore-Sigma, St. Louis, MO) was diluted in saline (50 µM), which was perfused into the head capsule through a T-tube switch. Repeated stimulation with Oct started 15 min after perfusion; by then the slowing-down of spiking activities were often noticeable. Care was taken not to introduce any air bubble into the tubing when switching from the syringe containing saline to the syringe containing the yohimbine solution. The water level in the two syringes was intentionally kept the

same in order to maintain a similar perfusing rate upon switching. The drug solution was kept flowing through the honey bee preparation until the end of protocol, which usually lasted for about 2 hr. No saline wash was attempted in this protocol due to the long time required for the recording sessions.

## Sequencing the *Amtyr1* region of the genome

We studied SNPs in full-genome sequences of eight *A. mellifera* workers (four high pollen hording and four low pollen hording). For each individual, Illumina short reads were mapped against the *A. mellifera* genome assembly version 4.5 (*Munoz-Torres et al., 2011*) using bwa version 0.5.9-r16 (*Li and Durbin, 2009*). An average 25× genome coverage per individual allowed the identification of high-quality SNPs in each individual against the reference genome. SNPs were identified with SAMtools version 0.1.17-r973:277 (*Li et al., 2009*) enforcing a minimum quality score of 20 (base call accuracy ≥99%).

## Statistical analysis

To analyze the effects in behavioral experiments, we used a generalized linear model with binomial error distribution and logit transformation to perform a logistic regression. The response variable is binomal (0,1). Trial is an ordered variable. We were most interested in testing the hypothesis that injection of yohimbine and dsiRNA before familiarization treatment would impact latent inhibition, so we focused on the interactions between trial, injection, injection time (before preexposure or before acquisition), and odor (novel or preexposed odor). To explore significant interactions further, we performed a tukey post hoc test using the package emmeans. All analyses were performed in R version 4.2.0 using RStudio version 2022.07.1.

## Acknowledgements

National Institutes of Health NIGMS (R01GM113967), Brian H Smith. National Science Foundation CRCNS (2113179), Brian H Smith. National Science Foundation NeuroNex (1559632), Brian H Smith co-PI. National Science Foundation BRAID (2223839), Brian H Smith co-PI. Department of Energy (SC0021922), Brian H Smith. The funders had no role in study design, data collection, and interpretation, or the decision to submit the work for publication.

## Additional information

### Funding

| Funder | Grant reference number | Author |
| --- | --- | --- |
| National Institutes of Health | R01GM113967 | Brian Smith |
| National Science Foundation | 2113179 | Brian Smith |
| National Science Foundation | 1559632 | Brian Smith |
| Department of Energy | SC0021922 | Brian Smith |

The funders had no role in study design, data collection, and interpretation, or the decision to submit the work for publication.

### Author contributions

Joseph S Latshaw, Xiaojiao Guo, Investigation, Methodology; Reece E Mazade, Mary Petersen, Investigation; Julie A Mustard, Supervision, Investigation, Methodology; Irina Sinakevitch, Formal analysis, Supervision, Validation, Investigation, Visualization, Methodology; Lothar Wissler, Methodology; Chelsea Cook, Software, Formal analysis; Hong Lei, Data curation, Formal analysis, Supervision, Validation, Investigation, Visualization, Methodology; Jürgen Gadau, Data curation, Formal analysis, Supervision, Validation, Investigation, Methodology; Brian Smith, Conceptualization, Investigation, Methodology, Supervision, Validation, Writing – original draft, Writing – review and editing

**Author ORCIDs**
Julie A Mustard https://orcid.org/0000-0002-1412-1140
Brian Smith https://orcid.org/0000-0001-7018-8561

## Decision letter and Author response
Decision letter https://doi.org/10.7554/eLife.83348.sa1
Author response https://doi.org/10.7554/eLife.83348.sa2

---

## Additional files

### Supplementary files
• MDAR checklist

### Data availability
All genomic, behavioral, electrophysiological and hplc data are available via Dryad.

The following datasets were generated:

| Author(s) | Year | Dataset title | Dataset URL | Database and Identifier |
|---|---|---|---|---|
| Smith B | 2023 | Tyramine and its AmTYR1 receptor modulate attention in honey bees (*Apis mellifera*) | https://doi.org/10.5061/dryad.gqnk98svb | Dryad Digital Repository, 10.5061/dryad.gqnk98svb |
| Gadau J, Smith B, Mustard JA, Sinakewitch I, Cook C, Lei H | 2023 | A tyramine receptor (*Amtyr1*) modulates attention in honey bees (*Apis mellifera*): A neural model for behavioral pleiotropy - Amtyr1-SNP genotypes | https://doi.org/10.5061/dryad.gqnk98svb | Dryad Digital Repository, 10.5061/dryad.prr4xgxsg |

---

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
