## [Editor Report]

This article reports a significant discovery: disrupting the function of the tyramine receptor in honey bees causes a rapid decline in their responses to olfactory stimuli. This finding highlights the important role of tyramine receptors, one of the most highly expressed biogenic amine receptors in the insect olfactory system. The authors propose that tyramine signaling may specifically control the process of latent inhibition, but the evidence presented does not rule out the possibility that tyramine affects other functions of the antennal lobe.

---

## [Decision Letter]

**Decision letter after peer review:**

Thank you for submitting your article "Tyramine and its AmTYR1 receptor modulate attention in honey bees (*Apis mellifera*)" for consideration by *eLife*.

Your article has been reviewed by three peer reviewers, and the evaluation has been overseen by Matthieu Louis as Reviewing Editor and Christian Rutz as Senior Editor. The following individuals involved in the review of your submission have agreed to reveal their identity: Alison Mercer (Reviewer #1); Wolfgang Blenau (Reviewer #2).

The reviewers have discussed their reviews with one another, and the Reviewing Editor has drafted this decision letter to help you prepare a revised submission.

Essential revisions:

All reviewers recognize the importance of the findings reported in the manuscript. Some concerns were raised about the interpretation of behavioral and physiological results exclusively through the lens of latent inhibition. It is possible that tyramine controls lateral inhibition in the antennal lobe without necessarily producing latent inhibition. Stating that a disruption of AmTyr1 increases the expression of latent inhibition without affecting appetitive learning (see abstract) does not appear to be fully supported by existing data. The fact that treatment with yohimbine produces a near "anti-bias" against the novelty odor (Figure 5D) highlights the complexity of the underlying regulatory mechanisms. Even if no link can be demonstrated between the function of AmTyr1 and latent inhibition, the results of the manuscript would remain very impactful due to the insight they provide into the potential function of AmTyr1 in the antennal lobe. It is nonetheless essential to address this shortcoming of the manuscript prior to publication.

Two alternatives are envisioned. First, the authors could provide additional experimental evidence supporting the specificity of the effects of AmTyr1 on latent inhibition with limited impact on appetitive learning. This could be achieved, for instance, by testing the effects of yohimbine on a differential conditioning paradigm to evaluate the ability of bees to discriminate between odors upon loss of function of AmTyr1. This experiment is just a suggestion. Other experiments might address this point equally well (or better). The reviewers fully appreciate the amount of work behind the present version of the manuscript. The authors are not expected to add a large body of new experimental data as long as they strengthen their conclusions related to latent inhibition. In the second alternative, the authors could modify their manuscript to acknowledge that the AmTyr1 pathway might regulate lateral inhibition without necessarily affecting latent inhibition. As part of this revision, it would be useful to present a model outlining potential roles for AmTyr1 based on the data presented in this manuscript and elsewhere.

In their revised manuscript, the authors are asked to address the comments of the reviewers about the statistics. They should also correct the color legend of Figure 3.

*Reviewer #1 (Recommendations for the authors):*

General comments: It may be helpful to view these results through the lens of lateral inhibition rather than latent inhibition. Please be clear also about how you interpret the results of your statistical analyses.

P3, L54: reduced? – Figure 1C suggests subsequent learning is delayed rather than reduced.

P5,105: reflex? – PE response.

P6: Is AmTYR1 the only tyramine receptor expressed in the honey bee brain?

Figure 3A: Use of the terms 'familiar' and 'novel' to refer to the two odours tested here is confusing. It would help to clarify on the figure also, that familiarization here is to air, not odour.

P8, L171-2, 179: The legend of Figure 3 suggests saline-treated bees are represented in blue and yohimbine-treated in orange. The opposite is suggested by the key in the figures.

P8, L180: 'familiarized to odor each of the two…… was familiarized..' – sentence?

P9, L190: If the time of year may be important, please specify.

P9, L195: Please explain the rationale here. How do puffs of air impact olfactory information processing in the antennal lobe? For example, do they engage in inhibitory networks?

Figure 3A: If yohimbine is injected immediately prior to conditioning, is there any change in the level of responses to odours? This seems important to test because in Figure 3B yohimbine treatment reduces levels of conditioned responses, whereas in Figure 3C learning in yohimbine-treated animals appears to be completely blocked.

P9, L212: "Injection of yohimbine eliminated the difference in responses to the novel and familiar odors" – yes, but it did so by reducing the responses to the novel odour. Given the variability in levels of responses in controls, it is difficult to determine whether what we see in Figure 3B, C is latent inhibition, or not. In 3B, for example, the acquisition rate of the familiar odour seems to be similar to that of the novel odour. In Figure 3C, there are too few trials to judge. Figure 1B suggests more than 4 trials may be required to show latent inhibition (as opposed to general inhibition) clearly (e.g. compare Figure 3BC; Figure 4B).

P10, L215-218: "This pattern could not arise from blockade of excitatory learning about N [the novel] odor because excitatory learning was unaffected in the air preexposure controls". Is it not more likely that puffs of air do not engage inhibitory circuits in the AL in the same way as puffs of odour?

P10, L219: what evidence suggests blockade of AmTYR1 "increases latent inhibition"? It seems more likely that lateral inhibition is increased and affects all odours.

P10, L222-224 Treatments in this study were given before or after odour familiarization (i.e. prior to acquisition only). Farooqui et al. (2003) examined the effects of treatments given either, prior to acquisition or prior to memory recall. As results presented in Figure 3C (and 4B) suggest blockade of TYR1 inhibits learning of novel as well as familiarized odours, it seems important to provide or refer to evidence showing that OA signalling is not disrupted, either by yohimbine, or by AmTYR1 dsiRNA injection.

P11, L238: The title of Figure 4 is misleading. Neither latent inhibition nor learning is apparent in bees treated with Amtyr1 dsiRNA. Do you mean increased lateral inhibition?

P11, L243-4: A significant effect of injection (treatment) is reported (<0.01). Doesn't this indicate a significant difference overall between response levels in control bees versus bees in which AmTYR1 is knocked down? This would seem consistent with an overall increase in lateral inhibition.

P11, L252: "...a slight decrement in response rate….(p<<0.01….). – slight?

P11, L261: Why was it predicted that disruption of AmTYR1 would attenuate latent inhibition?

P13, Figure 5 title: – what evidence is there here of latent inhibition? Presentation of results described in Figure 5 earlier might help clarify at the outset the magnitude and global nature of the changes induced by compromising the AmTYR1 function. This seems consistent also with increased lateral inhibition, rather than increased latent inhibition.

The results are fascinating. They suggest to me that tyramine is released tonically in the AL providing an essential brake on lateral inhibition.

*Reviewer #2 (Recommendations for the authors):*

Line 70ff "… a major locus … maps to a location in the honey bee genome…": I wonder if reference 21 is the correct reference at this point. Scheiner et al. did not carry out a mapping study but found a splice variant of the AmTYR1 receptor. Perhaps reference 24 (Page et al., 2000) would be more appropriate here?

Lines 124-139: This short half-page within the Results section does not refer to the results of the present study but essentially summarizes existing knowledge on the AmTYR1 receptor and VUM neurons in honey bees and other insects. The authors should consider presenting this information in the introduction.

Lines 147/147 "… the tyramine receptor antagonist yohimbine …": Although widely used in insect studies as a tyramine receptor antagonist, the specificity of yohimbine is not absolute. For example, yohimbine is also a high-affinity antagonist of the recently described honey bee α2-adrenergic-like octopamine receptor (reference 31). Yohimbine also has an antagonistic effect on the second tyramine receptor of the honey bee (Reim et al., 2017). From my point of view, this underlines the importance of the use of an alternative approach (DsiAmTyr1 treatment) by the authors. Unfortunately, no more specific antagonists are available either. Nevertheless, in my view, it would be best to point out this possible specificity issue.

Line 562ff "…primers for quantitative real-time PCR …": Can the authors please justify the choice of actin as the reference gene for qPCR? How has the stability of expression of the reference gene been checked?

*Reviewer #3 (Recommendations for the authors):*

All of my suggestions would be outside the scope of the paper.

I would enjoy seeing profiling of classical hPTMs associated with enhancer and regulatory sites (k27ac, K4m1, K4m3) via ChIP-seq, as well as associated RNA-seq analyses between the different lines or individuals showing variation in latent inhibition, in order to better understand the molecular components of this not directly relevant to the locus; however, this is a 'tall ask' for such a well-done paper.

The statistics were simple because this was appropriate.

For honeybee, the samples were well assessed, validated via dsiRNA and pharmacological methods, and interpretations were appropriately leveraged in light of the data.

I got nothing bar a bunch of genomics that aren't necessary…

[Editors’ note: further revisions were suggested prior to acceptance, as described below.]

Thank you for resubmitting your work entitled "Tyramine and its AmTYR1 receptor modulate attention in honey bees (*Apis mellifera*)" for further consideration by *eLife*. Your revised article has been evaluated by Christian Rutz (Senior Editor) and Matthieu Louis (Reviewing Editor).

The manuscript has been improved but there are some remaining issues that need to be addressed. Two of the initial three reviewers carefully went through the changes you implemented; the third reviewer was not available anymore. While we remain convinced that the results presented in this manuscript are fundamentally important, we also believe that the interpretation of your experimental data ought to be open to alternative explanations. The main problem is that the conclusions of the work rely on the use of indirect measures of latent inhibition. Presently, the electrophysiology results presented in Figure 5 fall short of unambiguously supporting latent inhibition. It is also difficult to apply the model proposed in Figure 7 to explain the results of Figure 5 (there is a concern about the fact that the original model of Ramaswami and colleagues has been partly distorted). Overall, the conceptual model of Figure 7 appears to bring more confusion than clarifications. These issues should be addressed prior to the publication of the work. We invite you to revise the manuscript along the lines suggested by the two reviewers -- please see their individual reports below.

We agree with the reviewers that reaching a mechanistic understanding of the function of AmTYR1 in the antennal lobe would be beyond the reach of a single study. Given the limitations of the experimental data presented in the manuscript, we ask that you acknowledge the possibility of explanations different from pure latent inhibition in your discussion of the results. Moreover, we recommend the addition of a reciprocal treatment to complete the electrophysiology inspection of Figure 5 (see comments of Reviewer #2). This addition offers an experimentally testable prediction that can be made based on the latent-inhibition model that you are proposing.

*Reviewer #1 (Recommendations for the authors):*

1. The authors conclude that disruption of Amtyr1 signaling increases the expression of latent inhibition but has little effect on appetitive conditioning (Abstract L29), but neither conclusion is clearly supported by the results presented in Figures3 and 4. Disruption of AmTYR1 reduced response levels (including responses to the novel odor) severely. As a result, latent inhibition could not be evaluated.

2. Previous work from the Smith lab revealed that local neurons (LNs) in the antennal lobes of the bee express octopamine receptors. This elegant work led to their proposal that octopamine inhibits inhibitory LNs in the glomerular core (leading to disinhibition of PNs) and simultaneously blocks excitation in neighboring glomeruli. It seems likely this could interfere with the generation and resilience of latent inhibition. In the experiments outlined here, appetitive learning performance is used to provide an indirect measure of latent inhibition induced by odor familiarization, but one difficulty in using this approach is that stimulation of the antennae with sucrose activates the VUMmx1 neuron (Hammer 1993), which will increase octopamine levels in the antennal lobes. The effects of sensitization and appetitive conditioning are therefore superimposed on effects induced by familiarization. What do the authors predict the outcome of this would be?

3. The model provided in Figure 7 does not seem to represent well the results of the electrophysiological analysis in this study (Figure 5). Disruption of AmTYR1 signalling (in the absence of odor familiarization) caused a dramatic decline in responses to odorants (Figure 5A,B), and rather than promoting latent inhibition, yohimbine treatment decreased odor response biases, and blocked the ability to enhance existing odor biases using familiarization (Figure 5C,D). The model presented in Figure 7, however, predicts that latent inhibition should be strongest when AmTYR1 function is blocked (Figure 7B). Doesn't this suggest something other than latent inhibition might be responsible for the global inhibition observed?

4. The authors suggest that Hebbian plasticity underlying latent inhibition is responsible for observed declines in odor responses (L429-433). As a result of disruption of Amtyr1 signaling, Hebbian plasticity, they argue, could induce a signal strong enough to produce inhibition to odors other than the familiar odor. This is interesting, but the model presented in Figure 7 is confusing and tells us little about how this might occur. The schematic suggests NMDA-receptor signaling in glomerulus X (top) leads to NMDA receptor-signaling in glomerulus Y. However, in glomerulus Y there is likely to be relatively little ORN-mediated excitation of PNs (or LNs). Also, in the lower glomerulus (Y), the retrograde signal (green arrow) appears to go from LN to PN. How would this work?

5. The authors acknowledge clearly that their model is based on a model of habituation in the AL of *Drosophila*, proposed by Ramaswami and colleagues. However, the authors have made subtle changes to the schematic provided by Twick et al. (2014) that could lead to some confusion. For example, excitatory inputs from ORNs onto PNs and LNs are depicted in the fly model as being distant from the NMDA receptor-mediated signalling proposed to underpin habituation. These synapses appear adjacent to the shaded area, which I assume represents the glomerular core. In the schematic presented in Figure 7, these synapses lie within the shaded area, which I assume now represents the glomerulus as a whole (core plus outer cortex). These differences may seem minor, but they could be misleading.

6. To help explain why AmTYR1 dysfunction gives rise to a global decline in odor responses, it would be helpful to provide a summary of neural networks in the AL of the bee. An excellent schematic presented by Smith and colleagues in an earlier publication (Sinakovitch et al. 2017) would be extremely helpful here. I believe the Sinakovitch model would make it much easier to discuss the results of this study, and their relationship with the fly habituation model.

7. At present, the authors do not comment at all on the unique roles of various subpopulations of LN in the bee or their functional properties. This omission seems odd given the central role LNs play in latent inhibition. Consideration of the functional properties of LNs also suggests alternative explanations for the general decline in odor responses observed in this study -- for example, the potential involvement of homogeneous LNs. Activation of these neurons, which have widefield arborizations throughout the AL, would be predicted to induce lateral inhibition that could potentially provide gain control. This seems highly relevant here, because it would help prevent saturation from the strong inputs generated as a result of compromised Amtyr1 function. I strongly recommend the papers from Rachel Wilson's group on this topic (e.g., Olsen et al. 2010).

In summary, I feel the bulk of the evidence presented in this paper points to an alternative explanation for the dramatic reduction in responses to odors induced by Amtyr1 knockdown. The results could potentially be a consequence of Amtyr1 knockdown inducing large responses that cause saturation in the network. In the absence of familiarization this can be controlled by lateral inhibition, but the process of familiarization, rather than leading to latent inhibition, causes further saturation and as a result, destabilization, which causes profound inhibition of the neural network.

I hope the comments above will be helpful, as intended.

*Reviewer #2 (Recommendations for the authors):*

In the revised version of their manuscript, the authors have responded to many of the reviewers' comments. In particular, they modified the discussion of the data significantly. They introduced a new Figure 7 in order to make it easier for the reader to understand the complex model of the effects of AmTYR1 activation in the antennal lobe. However, I do not find the model very understandable, and, in particular, the results of the electrophysiological recordings shown in Figure 5 are not sufficiently addressed in this model. In an attempt to understand these relations better, I again looked closely at Figure 5C+D of the manuscript. The question I asked myself was what the outcome of a reciprocal experiment would be: How does the number of units (neurons) that are more responsive to octanone change when familiarization is done with octanone instead of hexanol? A drop from 39% to a smaller number would likely be expected. Is that true? In this case, what is the influence of yohimbine injection? If yohimbine prevents the latent inhibition effect, yohimbine injection should prevent a decrease in the number (below 39%) or possibly even cause the number of responding neurons to rise (above 39%). Can this be shown experimentally? Alternatively, does yohimbine lead to a decrease in the number of octanone-biased neurons also in this constellation? This would argue for yohimbine causing a general decrease in the response to odorants. Can these assumptions or the results of the corresponding experiment be reconciled with the model shown in Figure 7? What would the model imply in this case? I hope the authors find my above suggestion constructive and I look forward to their response.

Line 482 "this plasticity it would give rise to": Delete "it".

---

## [Author Response]

Essential revisions:All reviewers recognize the importance of the findings reported in the manuscript. Some concerns were raised about the interpretation of behavioral and physiological results exclusively through the lens of latent inhibition. It is possible that tyramine controls lateral inhibition in the antennal lobe without necessarily producing latent inhibition. Stating that a disruption of AmTyr1 increases the expression of latent inhibition without affecting appetitive learning (see abstract) does not appear to be fully supported by existing data. The fact that treatment with yohimbine produces a near "anti-bias" against the novelty odor (Figure 5D) highlights the complexity of the underlying regulatory mechanisms. Even if no link can be demonstrated between the function of AmTyr1 and latent inhibition, the results of the manuscript would remain very impactful due to the insight they provide into the potential function of AmTyr1 in the antennal lobe. It is nonetheless essential to address this shortcoming of the manuscript prior to publication.Two alternatives are envisioned. First, the authors could provide additional experimental evidence supporting the specificity of the effects of AmTyr1 on latent inhibition with limited impact on appetitive learning. This could be achieved, for instance, by testing the effects of yohimbine on a differential conditioning paradigm to evaluate the ability of bees to discriminate between odors upon loss of function of AmTyr1. This experiment is just a suggestion. Other experiments might address this point equally well (or better). The reviewers fully appreciate the amount of work behind the present version of the manuscript. The authors are not expected to add a large body of new experimental data as long as they strengthen their conclusions related to latent inhibition. In the second alternative, the authors could modify their manuscript to acknowledge that the AmTyr1 pathway might regulate lateral inhibition without necessarily affecting latent inhibition. As part of this revision, it would be useful to present a model outlining potential roles for AmTyr1 based on the data presented in this manuscript and elsewhere.

We have chosen the second alternative as the means to respond to this commentary. The now extensive modification, esp of the discussion, is highlighted in blue text in the revision and is summarized in a model presented in a new figure 7 in the Discussion. We chose not to use the first option. As we now review in the new paragraph two in the Discussion section, Amtyr1 has been shown to be involved in several different behaviors and physiological process in addition to latent inhibition. A new experiment, e.g. differential conditioning, might just add to this now lengthy list without providing more resolution as to how this gene is acting. We feel we have highlighted this issue now much better than before as a result of this reviewer’s comments.

In their revised manuscript, the authors are asked to address the comments of the reviewers about the statistics. They should also correct the color legend of Figure 3.

Color legend is corrected in Figure 3.

Reviewer #1 (Recommendations for the authors):General comments: It may be helpful to view these results through the lens of lateral inhibition rather than latent inhibition. Please be clear also about how you interpret the results of your statistical analyses.

In regard to this reviewer’s comments on weaknesses, and to this general one, we have made significant edits to the discussion as well as to a few other places in the manuscript. In sum, and as we now extensively elaborate on in the Discussion, we feel that part of the mechanism for latent inhibition is through its effect on Hebbian plasticity that modifies both feed-forward and lateral inhibition. In other words, latent inhibition and lateral/feed-forward inhibition is intertwined such that one cannot be discussed without the other.

P3, L54: reduced? – Figure 1C suggests subsequent learning is delayed rather than reduced.

This is in the introduction in reference to reduction of learning. This reviewer refers to patterns in Figure 1C that appears to show a delay as learning about the familiar odor in ‘Inhibitor’ drones at one end of the curve (blue) shown in Figure 1B lags behind learning about the familiar odor in ‘Noninhibitor’ (green) drones at the right of the distribution in Figure 1B. I would not dispute that learning in ‘inhibitor’ bees will ultimately reach levels of asymptotic responding in noninhibitor bees. But to me ‘reduced’ in the use here refers to any tendency for inhibitor bees to respond less than noninhibitor bees in rate of acquisition and/or in asymptotic levels of responding. And in fact sometimes we see slightly different patterns of responses to novel and familiar odors. That is evident in Figures3 and 4. But in all cases responding to the familiar odor is less than that to the novel odor. The differences in responses across experiments that are well separated in time could arise because of epigenetic modification of Amtyr1 action. That modification is something we will investigate, and will soon report on some of the first experiments in a subsequent manuscript currently in preparation. In the meantime, in an attempt to convey these subtle differences across experiments, I have modified the text on page 3 line 55 to read ‘…delayed or reduced…’. Also, just above that point in the text we refer to learning being ‘…delayed or slower…’

P5,105: reflex? – PE response.

Fixed.

P6: Is AmTYR1 the only tyramine receptor expressed in the honey bee brain?

Yes, there is one other tyramine receptor identified to date – *Amtyr2* (Reim et al. 2017. Insect Biochem Mol Biol 80: 91-100). For the reasons outlined here we do not go into detail about Amtyr2 in our manuscript, and we assume that these comments will be available to readers where published.

Expression of *Amtyr1* and *Amtyr2*: In the Reim et al. paper, the authors dissect out regions of the brain and use quantitative PCR to compare expression of each receptor between foragers and nurse bees. The results confirm the expression of both *Amtyr1* and *Amtyr2 in* the central brain, optic lobes, antennal lobes and subesophageal ganglion. These results are shown in their supplemental data (Figure S1). They did not compare the expression levels of *Amtyr1* to *Amtyr2*, they just compared nurses to foragers.

Since there has not been any work done looking at *Amtyr2* expression in sections or using in situ or immunohistochemistry, as we have done for Amtyr1, we do not know what specific cells express AmTYR2 beyond knowing it is expressed in those general regions.

Conclusion: *Amtyr1* and *Amtyr2* are both expressed throughout the brain, and we currently do not know if their expression patterns overlap.

2) The *Amtyr1* and *Amtyr2* genes are both located on linkage group 1 (chromosome 1). However, they are located 10.3 Mbp (million base pairs) away from each other. The 10 centiMorgan region surrounding *Amtyr1* shown in figure 2B in our manuscript encompasses 0.28 Mbp. There are likely to be multiple RAPD and SNP markers between the *Amtyr1* and *Amtyr2* genes (est941, est294, etc.), which would have separated their effects in the QTL mapping.

Conclusion: Although both tyramine receptor genes are on chromosome 1, the *Amtyr2* gene is far away from the identified QTL that includes *Amtyr1* and is unlikely to have contributed to the effect.

3) Both Amtyr1 and Amtyr2 are blocked by yohimbine. However, our RNA probes were designed to be specific for Amtyr1 (see Guo et al. 2018. J Insect Physiol 111: 47-52).

Figure 3A: Use of the terms 'familiar' and 'novel' to refer to the two odours tested here is confusing. It would help to clarify on the figure also, that familiarization here is to air, not odour.

We agree that it is confusing. Figures 3 and 4 now refer both in the name of the odors and symbols to represent that they are both novel given that air used was used for familiarization.

P8, L171-2, 179: The legend of Figure 3 suggests saline-treated bees are represented in blue and yohimbine-treated in orange. The opposite is suggested by the key in the figures.

Fixed in the figure legend.

P8, L180: 'familiarized to odor each of the two…… was familiarized..' – sentence?

Corrected.

P9, L190: If the time of year may be important, please specify.

Time of year is one of several variables that affect performance in PER conditioning. As noted in that section of text, timing of injections in terms of proximity to training has an effect to. Because these variables were randomized out across saline and yohimbine injection groups, it is difficult to get into too much detail about what may, or may not, have had an effect. That is, other than mentioning it in the text. Also, effects of season are bound to be very different in desert conditions in Arizona versus more temperate or tropical climates.

P9, L195: Please explain the rationale here. How do puffs of air impact olfactory information processing in the antennal lobe? For example, do they engage in inhibitory networks?

We now discuss in the revision (pg. 19 around line 450) how mechanosensory stimulation with air affects antennal lobe processing, and how it is different from air containing odor.

Figure 3A: If yohimbine is injected immediately prior to conditioning, is there any change in the level of responses to odours? This seems important to test because in Figure 3B yohimbine treatment reduces levels of conditioned responses, whereas in Figure 3C learning in yohimbine-treated animals appears to be completely blocked.

We don’t know the answer to this question. It is possible that there is some reduction in responsiveness either to yohimbine and/or to the injection itself directly prior to conditioning. We thought it important to show with the experiment in 3C that injection timing was not critical to the qualitative results shown in 3B – that is that yohimbine led to an overall reduction in responsiveness and to no difference in responding to Novel and Familiar odors. In the end, the experiment should in 3C is not critical and can be deleted, in particular because we do not have comparable data in Figure 4 for reasons cited in the text.

P9, L212: "Injection of yohimbine eliminated the difference in responses to the novel and familiar odors" – yes, but it did so by reducing the responses to the novel odour. Given the variability in levels of responses in controls, it is difficult to determine whether what we see in Figure 3B, C is latent inhibition, or not. In 3B, for example, the acquisition rate of the familiar odour seems to be similar to that of the novel odour. In Figure 3C, there are too few trials to judge. Figure 1B suggests more than 4 trials may be required to show latent inhibition (as opposed to general inhibition) clearly (e.g. compare Figure 3BC; Figure 4B).

As with all PER studies, there is variability from experiment-to-experiment in how well any form of learning is expressed. In our studies, every experiment in this and other publications (cited in the manuscript) show lower responsiveness to the familiar odor than to the novel. Sometimes this shows up in initial responses, rate of acquisition or asymptotic responses. There is also visual component in addition to the odor specific component, which means the behavior is far more complicated than it looks. In fact, as we review here, much individual variability arises from genetic differences that, as we have shown elsewhere, are important for colony fitness. So some individual bees, and bees from lines selected for low latent inhibition, don’t show the behavior at all.

Here we selected a mix of bees, some of which would show strong latent inhibition and some weak or not at all. We did that, as we describe in the manuscript, so that latent inhibition could be increased or decreased from that mixed baseline of bees. But that means that as we select a small sample from that mixed baseline – ~20-25 bees – that we will sometimes get groups somewhat biased toward showing it or not. That is certainly one of the reasons why there is variation from experiment to experiment. But this variation does not undermine the basic fact that response levels to familiar odors is lower than to novel, one way or another.

P10, L215-218: "This pattern could not arise from blockade of excitatory learning about N [the novel] odor because excitatory learning was unaffected in the air preexposure controls". Is it not more likely that puffs of air do not engage inhibitory circuits in the AL in the same way as puffs of odour?

Yes, we agree. We now describe in detail in the model presented in the new figure 7 that odor itself is much more salient and that it triggers more and different activity in the antennal lobe. All of this is discussed in the now heavily rewritten discussion. As noted in the manuscript, as well as in the responses in this document, lateral (and feedforward) inhibition is likely an important physiological mechanism that underlies latent inhibition to odor. But odor is clearly the more salient stimulus.

P10, L219: what evidence suggests blockade of AmTYR1 "increases latent inhibition"? It seems more likely that lateral inhibition is increased and affects all odours.

The evidence, it seems to us, is that the increase in inhibition is specific to odor ‘familiarization’. That pre-exposure to odor induces behavioral latent inhibition – poor learning of the familiar odor and significantly better learning of the novel odor. The effect of tyramine disruption is then specific to treatments that have odor presentation during the familiarization phase.

If tyramine blockade just increased lateral inhibition, then why is it not present to the same degree after air familiarization as it is when odor is preexposed in the familiarization phase? It seems clear that there is some effect of habituation to odor – the most salient part of the stimulus delivery – that is modulated by tyramine. Note that there is a slight effect of air in Figure 4A, which we discuss at the end of the discussion. But it is quantitatively and qualitatively different for when odor is preexposed.

We now present in Figure 7 a model that encompasses the relationship between inhibitory processes in the antennal lobe, which may also cover other areas of the brain, and the generation of odor-driven latent inhibition.

This is not to say that we have solved all the outstanding questions. There are still many issues to work out in much more detail. But we feel that our contribution at least reveals a very interesting and novel problem that needs much more attention.

P10, L222-224 Treatments in this study were given before or after odour familiarization (i.e. prior to acquisition only). Farooqui et al. (2003) examined the effects of treatments given either, prior to acquisition or prior to memory recall. As results presented in Figure 3C (and 4B) suggest blockade of TYR1 inhibits learning of novel as well as familiarized odours, it seems important to provide or refer to evidence showing that OA signalling is not disrupted, either by yohimbine, or by AmTYR1 dsiRNA injection.

Our treatments are also prior to acquisition and prior to recall, as in Farooqui et al. But keep in mind that the acquisition for latent inhibition is the odor familiarization phase, when odor is presented without reinforcement. The phase in which odors are reinforced is the test phase.

It is likely that yohimbine affects octopamine receptors. But it is much less likely that dsRNA would do that. And if it did, overall learning would be reduced – à la Farooqui et al. – even after air preexposure, and it would not be specific to a familiar odor. Moreover, there are no octopamine receptors within the confidence limits of the locus that came up twice in the genetic studies.

But, given the complexities of the neural circuitry, it may well be that other receptors are involved. So, while we can conclude that Amtyr1 is involved in latent inhibition via its effect through lateral inhibition, we cannot rule out the possible roles of other receptors. To work out first what those receptors are, and then experimentally manipulate them, goes well beyond what can be addressed in a single publication. And that would probably require use of the fruit fly model, as suggested by Reviewer #2 and as listed in the box now in the discussion.

P11, L238: The title of Figure 4 is misleading. Neither latent inhibition nor learning is apparent in bees treated with Amtyr1 dsiRNA. Do you mean increased lateral inhibition?

Actually, learning is apparent – significant trial effect without significant interaction term, as reported in the legend – in Amtyr1 dsRNA treated bees in Figure 4 A. As in our response above, the big ‘inhibitory’ effect of Amtyr1 blockade is specific to when odor is preexposed, in Figure 4B, which gives rise to a significant interaction term.

P11, L243-4: A significant effect of injection (treatment) is reported (<0.01). Doesn't this indicate a significant difference overall between response levels in control bees versus bees in which AmTYR1 is knocked down? This would seem consistent with an overall increase in lateral inhibition.

Yes, which we discuss in the heavily re-written Discussion section. But this general increase would not easily account for the qualitative differences between air familiarization (4A) and odor familiarization (4B).

P11, L252: "..a slight decrement in response rate….(p<<0.01….). – slight?

The word ‘slight’ has been deleted. It is significant, and that effect is discussed in the Discussion (pg 19, line 452-455) as potentially being important for understanding the mechanism. But the difference between 4A and 4B is in the significant interaction term in the latter but not the former. That term shows the effect of odor familiarization and implies the Hebbian mechanism that we have now put into the model in Figure 7.

P11, L261: Why was it predicted that disruption of AmTYR1 would attenuate latent inhibition?

This section was written to convey our surprise that the effect was not a standard effect for attempts to disrupt learning (e.g. dunce, rutabaga, amnesiac – which all block learning in fruit flies). But we clearly anticipated that this effect *could be* in the direction we observed because we used mixed genotypes in the learning studies.

P13, Figure 5 title: – what evidence is there here of latent inhibition? Presentation of results described in Figure 5 earlier might help clarify at the outset the magnitude and global nature of the changes induced by compromising the AmTYR1 function. This seems consistent also with increased lateral inhibition, rather than increased latent inhibition.

In retrospect, the question of how latent inhibition is shown in this figure may have been less than clear. We have now modified the text in the section starting around line 281, which describes the experiments more clearly. In particular, we clarify how Figures5A and 5B shows the link between Amtyr1 and inhibition. We also clarify how %c and %D show disruption of latent inhibition via its manifestation of novelty detection.

The results are fascinating. They suggest to me that tyramine is released tonically in the AL providing an essential brake on lateral inhibition.

The model we now propose in Figure 7 describes how we think tyramine provides this brake. And we try to clarify that we think lateral (and feedforward) inhibition is critical to production of latent inhibition.

Reviewer #2 (Recommendations for the authors):Line 70ff "… a major locus … maps to a location in the honey bee genome…": I wonder if reference 21 is the correct reference at this point. Scheiner et al. did not carry out a mapping study but found a splice variant of the AmTYR1 receptor. Perhaps reference 24 (Page et al., 2000) would be more appropriate here?

Fixed. Ref 21 is now Page et al. 2000.

Lines 124-139: This short half-page within the Results section does not refer to the results of the present study but essentially summarizes existing knowledge on the AmTYR1 receptor and VUM neurons in honey bees and other insects. The authors should consider presenting this information in the introduction.

We prefer to keep this in place. In doing so it reflects more the process of how we came to focus on Amtyr1 in this genetic region, that is, over any other gene in this region. To put it in the intro might imply that we anticipated Amtyr1 coming out of the genetic screen.

Lines 147/147 "… the tyramine receptor antagonist yohimbine …": Although widely used in insect studies as a tyramine receptor antagonist, the specificity of yohimbine is not absolute. For example, yohimbine is also a high-affinity antagonist of the recently described honey bee α2-adrenergic-like octopamine receptor (reference 31). Yohimbine also has an antagonistic effect on the second tyramine receptor of the honey bee (Reim et al., 2017). From my point of view, this underlines the importance of the use of an alternative approach (DsiAmTyr1 treatment) by the authors. Unfortunately, no more specific antagonists are available either. Nevertheless, in my view, it would be best to point out this possible specificity issue.

So noted in text around lines 237-239.

Line 562ff "…primers for quantitative real-time PCR …": Can the authors please justify the choice of actin as the reference gene for qPCR? How has the stability of expression of the reference gene been checked?

I have added a new reference (75) discussing the use of actin as a control for many different siRNA manipulations. Also, this experiment used a scrambled Amtyr1 sequence as the control, and the result was that expression of Amtyr1 was reduced three-fold relative to the scrambled sequence. Given no change in actin, and reduction relative to scrambled, we feel that this experiment was well controlled. Checking for stability, as we understand this comment, would require one or two more reference genes in addition to actin.

Reviewer #3 (Recommendations for the authors):All of my suggestions would be outside the scope of the paper.I would enjoy seeing profiling of classical hPTMs associated with enhancer and regulatory sites (k27ac, K4m1, K4m3) via ChIP-seq, as well as associated RNA-seq analyses between the different lines or individuals showing variation in latent inhibition, in order to better understand the molecular components of this not directly relevant to the locus; however, this is a 'tall ask' for such a well-done paper.

This manuscript, as reference above, will be forthcoming very soon as a follow up to this publication.

The statistics were simple because this was appropriate.For honeybee, the samples were well assessed, validated via dsiRNA and pharmacological methods, and interpretations were appropriately leveraged in light of the data.I got nothing bar a bunch of genomics that aren't necessary…

We removed one figure describing locations of snp’s in and around the Amtyr1 locus. This was not too informative, as this reviewer suggests. We also moved the text on this figure to the section describing the QTL (around line 145).

[Editors’ note: what follows is the authors’ response to the second round of review.]

The manuscript has been improved but there are some remaining issues that need to be addressed. Two of the initial three reviewers carefully went through the changes you implemented; the third reviewer was not available anymore. While we remain convinced that the results presented in this manuscript are fundamentally important, we also believe that the interpretation of your experimental data ought to be open to alternative explanations. The main problem is that the conclusions of the work rely on the use of indirect measures of latent inhibition. Presently, the electrophysiology results presented in Figure 5 fall short of unambiguously supporting latent inhibition. It is also difficult to apply the model proposed in Figure 7 to explain the results of Figure 5 (there is a concern about the fact that the original model of Ramaswami and colleagues has been partly distorted). Overall, the conceptual model of Figure 7 appears to bring more confusion than clarifications. These issues should be addressed prior to the publication of the work. We invite you to revise the manuscript along the lines suggested by the two reviewers -- please see their individual reports below.We agree with the reviewers that reaching a mechanistic understanding of the function of AmTYR1 in the antennal lobe would be beyond the reach of a single study. Given the limitations of the experimental data presented in the manuscript, we ask that you acknowledge the possibility of explanations different from pure latent inhibition in your discussion of the results. Moreover, we recommend the addition of a reciprocal treatment to complete the electrophysiology inspection of Figure 5 (see comments of Reviewer #2). This addition offers an experimentally testable prediction that can be made based on the latent-inhibition model that you are proposing.

I asked a colleague who works in behavior genetics to review the manuscript. He pointed out that the outcome of our experiments is not what one would expect from a typical knockout experiment. I agree and have pointed that out in the manuscript’s discussion and in the response letters. But readers may approach the manuscript with that strong expectation, which is what is leading to some of the reviews. He convinced me it would be good to revise the introduction to set readers up for a story that is very different from what one expects. I have added the last paragraph to the introduction in an effort to describe what the ‘counterintuitive’ outcome of the story will be. I hope this clarifies a story that is, admittedly, more complicated that one normally expects.

In regard to the reciprocal experiment in Figure 5, we have done that and reported the fully counterbalanced results in an earlier publication (Lei et al. #51) cited in the manuscript.

Reviewer #1 (Recommendations for the authors):1. The authors conclude that disruption of Amtyr1 signaling increases the expression of latent inhibition but has little effect on appetitive conditioning (Abstract L29), but neither conclusion is clearly supported by the results presented in Figures3 and 4. Disruption of AmTYR1 reduced response levels (including responses to the novel odor) severely. As a result, latent inhibition could not be evaluated.

There is a misunderstanding here, which stems from the claim that “Disruption of AmTYR1 reduced response levels (including responses to the novel odor) severely. As a result, latent inhibition could not be evaluated.” This statement is incorrect. We show that blockade (Figure 3A) or disruption (Figure 4A) of amtyr1 has no or only slight effects on excitatory conditioning. But the same treatment that produces behavioral latent inhibition – odor familiarization – in control animals (saline or scrambled dsRNA injected animals) produces a profound reduction in odor responsiveness in treatment animals (yohimbine or dsRNA groups). The reduction is specific to the familiarization treatment that produces latent inhibition (in Figure 3B,C but not 3A, and in Figure 4B but not 4A). This indicates that amtyr1 could be affecting a neural mechanism that somehow modulates latent inhibition.

What we have done is a standard protocol for studying the effects of any gene on a behavior. One disrupts the gene (or its product) and shows that the behavior is disrupted. Therefore, it does not seem reasonable to us to conclude that amtyr1’s role in latent inhibition could not be evaluated because, when it is disrupted, animals no longer show latent inhibition.

The question is how this disruption occurs. Part of the misunderstanding is that latent inhibition is disrupted in an unexpected and counterintuitive way, which in fact we did not anticipate when we ran the experiments (and which is described in the Discussion in paragraph starting around line 443). As noted, normally one establishes a behavior and then tries to disrupt a neural pathway that is associated with the behavior. In this case, we anticipated that upon disruption of amtyr1 the response to the familiar odor would rise to equal response to the novel odor, which would remain unaffected.

But that did not happen either with pharmacological or dsRNA treatments. Instead, blockade or disruption of amtyr1 drags down the response to the novel odor to be equal to that for the familiar odor. For that reason we do not reach the conclusion that amtyr1 produces latent inhibition*.* But amtyr1 clearly affects expression of latent inhibition.

We try to be clear throughout the manuscript (and in the title) that it *modulates* expression of latent inhibition, and we ultimately argue that it does so by *indirectly* modifying Hebbian plasticity in inhibitory networks. (This mechanism is not unlike what this reviewer describes). Indeed in figure 7 we show based on our published data (Sinakevitch et al. 2017) that amtyr1 is operating at synapses separate from, and upstream of, the synapses that we and Ramaswami et al. have proposed for Hebbian-plasticity dependent latent inhibition.

The sentence this reviewer refers to in the abstract is: “…We then show that disruption of *Amtyr1* signaling either pharmacologically or through RNAi increases expression of latent inhibition but has little effect on appetitive conditioning, and these results suggest that AmTYR1 modulates inhibitory processing in the CNS.” The highlighted text now reads: “…qualitatively changes expression of latent inhibition…”. But note that even in the original version we write that it changes *expression* of latent inhibition, not that it disrupts latent inhibition. We also write in that sentence that it modulates inhibitory processing.

In addition, in the first revision we added the second paragraph in the Discussion in which we explicitly write “…it [amtyr1] is not a latent inhibition gene’ (Lines 385-394). Importantly, our interpretation of the effect of amtyr1 – that it is modulating sensory inputs – can potentially explain the broader pleiotropic effects that have been documented for amtyr1. This point is made in the Discussion.

Admittedly, some of the confusion may stem from the way we described the behavior in prior versions of this manuscript, some of which made it into these later versions. But our thinking has evolved, in part as a result of these very thorough reviews. We have now made new changes throughout the manuscript to hopefully remove any ambiguity in regard to this issue. It needs to be clear that the modulation by amtyr1 – at afferent axon terminals in the antennal lobe and mushroom body – is separate from the familiarization-induced mechanism for latent inhibition at LN-to-PN or PN-to-MB terminals.

2. Previous work from the Smith lab revealed that local neurons (LNs) in the antennal lobes of the bee express octopamine receptors. This elegant work led to their proposal that octopamine inhibits inhibitory LNs in the glomerular core (leading to disinhibition of PNs) and simultaneously blocks excitation in neighboring glomeruli. It seems likely this could interfere with the generation and resilience of latent inhibition. In the experiments outlined here, appetitive learning performance is used to provide an indirect measure of latent inhibition induced by odor familiarization, but one difficulty in using this approach is that stimulation of the antennae with sucrose activates the VUMmx1 neuron (Hammer 1993), which will increase octopamine levels in the antennal lobes. The effects of sensitization and appetitive conditioning are therefore superimposed on effects induced by familiarization. What do the authors predict the outcome of this would be?

The measure we use for Latent Inhibition – retardation of acquisition – is the standard *direct* behavioral measure for latent inhibition (see refs 4 and 5 by Lubow). Unreinforced presentation of any CS sets up non associative plasticity, the memory of which interferes with the mechanisms that produce and/or express excitatory conditioning, as this reviewer’s comment suggests for octopamine. To rephrase this reviewer’s comment, latent inhibition likely interferes with the generation and resilience of excitatory conditioning, which is the basic mechanism of latent inhibition implied by retardation of acquisition. We have shown retardation of acquisition in detail in Chandra et al. (cited ref #6), including failure of summation, which is the other direct measure of latent inhibition.

In fact, in an earlier publication my colleagues and I provided a framework for how this interaction could take place. We show via computational modeling how neural mechanisms of non associative (latent inhibition), associative and operant conditioning could interact to shape the form of behavioral responses:

A computational framework for understanding decision making through integration of basic learning rules. Bazhenov M, Huerta R, Smith BH. J Neurosci. 2013 Mar 27;33(13):5686-97. doi: 10.1523/JNEUROSCI.4145-12.2013. PMID: 23536082

And we have models for octopamine action in associative conditioning in the antennal lobe that could be adapted here, e.g.:

Learning modifies odor mixture processing to improve detection of relevant components.

Chen JY, Marachlian E, Assisi C, Huerta R, Smith BH, Locatelli F, Bazhenov M. J Neurosci. 2015 Jan 7;35(1):179-97. doi: 10.1523/JNEUROSCI.2345-14.2015. PMID: 25568113

We certainly need to look at this interaction both computationally and experimentally. But this is thesis-level work that is beyond the scope of this publication.

3. The model provided in Figure 7 does not seem to represent well the results of the electrophysiological analysis in this study (Figure 5). Disruption of AmTYR1 signalling (in the absence of odor familiarization) caused a dramatic decline in responses to odorants (Figure 5A,B), and rather than promoting latent inhibition, yohimbine treatment decreased odor response biases, and blocked the ability to enhance existing odor biases using familiarization (Figure 5C,D). The model presented in Figure 7, however, predicts that latent inhibition should be strongest when AmTYR1 function is blocked (Figure 7B). Doesn't this suggest something other than latent inhibition might be responsible for the global inhibition observed?

The short answer to the question at the end is ‘no’. First, potentiation of the response to the novel odor after familiarization occurs via a mechanism we do not, and at this point for lack of data we cannot, represent in Figure 7. That potentiation could occur via a so far unknown mechanism intrinsic to the antennal lobe neural networks, or, perhaps more likely, it could occur via feedback from mechanisms in the mushroom bodies that are known to produce potentiation to novel stimuli (new ref #67). We only show in Figure 5 that expression of it depends on amtyr1 function.

Under normal circumstances amtyr1 is functional and moderating the Hebbian plasticity at LN-to-PN synapses at levels consistent with Figures7C or 7D, where novel odors are learned well (likely aided by potentiation). Blocking or disrupting amtyr1 puts the network in a state consistent with 7B, where responses to all odors are affected by very strong Hebbian plasticity at LN-to-PN synapses – including ‘potentiated’ novel odors given the overlap in sensory representations (see the response to the next comment [#4]). It is clear in the behavioral data that acquisition to novel odors is blocked by familiarization to odor with amtyr1 disruption. And this blockade does not occur in the controls when just air was used with the same amtyr1 disruption, which shows that it is not simply augmentation of inhibition outside of familiarization treatment.

We try to clarify this now with blue text paraphrased from this response in the Discussion of the manuscript (lines 525-545).

4. The authors suggest that Hebbian plasticity underlying latent inhibition is responsible for observed declines in odor responses (L429-433). As a result of disruption of Amtyr1 signaling, Hebbian plasticity, they argue, could induce a signal strong enough to produce inhibition to odors other than the familiar odor. This is interesting, but the model presented in Figure 7 is confusing and tells us little about how this might occur. The schematic suggests NMDA-receptor signaling in glomerulus X (top) leads to NMDA receptor-signaling in glomerulus Y. However, in glomerulus Y there is likely to be relatively little ORN-mediated excitation of PNs (or LNs). Also, in the lower glomerulus (Y), the retrograde signal (green arrow) appears to go from LN to PN. How would this work?

First, the arrow in glomerulus Y was incorrect. It should be flipped, and is correctly represented in the new version of the manuscript. Now it shows the retrograde signal going from PNy to the LN. I apologize for that error.

What this reviewer writes above is incorrect: “…in glomerulus Y there is likely to be relatively little ORN-mediated excitation of PNs (or LNs).” As we describe in lines 494 to 506 of the Discussion, and as has been described in several publications including in Figure 1 (see Figure A C-8 secondary ketone vs C-6 primary alcohol) of the Paoli and Galizia (2021) review we cite, there is overlap in odor activity maps for most of the common monomolecular odors used in behavioral and physiological studies of olfaction in honey bees, including the two odors we use in this study. So, under blockade or disruption of amtyr1, which would increase sensory drive from ORN axon terminals, Hebbian plasticity could easily occur in the representation for the novel odor under conditions of amtyr1 disruption.

5. The authors acknowledge clearly that their model is based on a model of habituation in the AL of *Drosophila*, proposed by Ramaswami and colleagues. However, the authors have made subtle changes to the schematic provided by Twick et al. (2014) that could lead to some confusion. For example, excitatory inputs from ORNs onto PNs and LNs are depicted in the fly model as being distant from the NMDA receptor-mediated signalling proposed to underpin habituation. These synapses appear adjacent to the shaded area, which I assume represents the glomerular core. In the schematic presented in Figure 7, these synapses lie within the shaded area, which I assume now represents the glomerulus as a whole (core plus outer cortex). These differences may seem minor, but they could be misleading.

We attempted to adopt as faithfully as possible the figure from Twick et al. The change in the shaded area was not meant to imply anything about where the synapses were made (inside or outside of a glomerulus core or cortex). This has now been clarified in the legend for this figure.

6. To help explain why AmTYR1 dysfunction gives rise to a global decline in odor responses, it would be helpful to provide a summary of neural networks in the AL of the bee. An excellent schematic presented by Smith and colleagues in an earlier publication (Sinakovitch et al. 2017) would be extremely helpful here. I believe the Sinakovitch model would make it much easier to discuss the results of this study, and their relationship with the fly habituation model.

See reply to point #7.

7. At present, the authors do not comment at all on the unique roles of various subpopulations of LN in the bee or their functional properties. This omission seems odd given the central role LNs play in latent inhibition. Consideration of the functional properties of LNs also suggests alternative explanations for the general decline in odor responses observed in this study -- for example, the potential involvement of homogeneous LNs. Activation of these neurons, which have widefield arborizations throughout the AL, would be predicted to induce lateral inhibition that could potentially provide gain control. This seems highly relevant here, because it would help prevent saturation from the strong inputs generated as a result of compromised Amtyr1 function. I strongly recommend the papers from Rachel Wilson's group on this topic (e.g., Olsen et al. 2010).

We present in Figure 7 a simplified, heuristic model aimed toward helping readers understand what we think is happening in the antennal lobe and possibly at PN axon synapses in the mushroom body. This model is not meant to represent the full complexity of neural networks at either level of processing. The sort of discussion to generate a full mechanistic understanding of what is happening with amtyr1 in both networks, involving all of the different cell types in either location, would require computational modeling to understand the complexity. That can and should be done, but it is beyond the scope of one publication.

Nevertheless, in response to this review we present some information along these lines. We feel that the most likely LNs represented in Figure 7 would be the heterogeneous LNs, as represented in a previous model of the antennal lobe published by my lab. We include a paragraph in the discussion on this topic (lines 507-524). In that paragraph we also reference a prior publication Sinakevitch et al. (2017) where we present a more detailed model of the antennal lobe circuitry including amtyr1 and amoa1, a receptor for octopamine thought to be involved in excitatory conditioning.

In summary, I feel the bulk of the evidence presented in this paper points to an alternative explanation for the dramatic reduction in responses to odors induced by Amtyr1 knockdown. The results could potentially be a consequence of Amtyr1 knockdown inducing large responses that cause saturation in the network. In the absence of familiarization this can be controlled by lateral inhibition, but the process of familiarization, rather than leading to latent inhibition, causes further saturation and as a result, destabilization, which causes profound inhibition of the neural network.

I have put new paragraphs in the manuscript to clarify our interpretations, which will require much more physiological-level analyses and computational modeling. These paragraphs include an elaboration of one already in the Discussion from the first revision on the likelihood that *behavioral* latent inhibition arises as a result of distributed neural mechanisms in at least the antennal lobes and mushroom bodies (see ref 67 and lines 401-417).

But the review here is forcing us to defend a position – i.e., that we can explain behavioral latent inhibition solely in neural networks of the antennal lobes – that is too premature and speculative, and is quite possibly wrong. Explaining behavior, after we treat the entire brain, by just focusing on the antennal lobes is too speculative. We have shown that Amtyr1 is expressed in both the antennal lobes and mushroom body on afferent axon terminals from ORNs and PNs. We actually discuss this issue in the third paragraph (lines 395-411) in the Discussion, which we added as a result of the previous review.

We are proposing a model, based on a request for one in the previous review, for encoding of a correlate of latent inhibition in the antennal lobes that makes sense based on our prior analyses and those of Ramaswami et al., and based on our behavioral and electrophysiological data in this manuscript. The correlation is defined as a neural effect that is produced by the same familiarization treatment that produces behavioral latent inhibition, and which is affected by blockade or disruption of amtyr1. We feel that what is happening in the antennal lobe contributes to the behavior, but it is likely functioning with at least a similar processing mechanism in the mushroom bodies. To go into too much detail about the neural networks of the antennal lobes at this point, beyond what we do to explain the neural mechanism as we see it, would distract from the broader point that this mechanism might be acting in the mushroom bodies too.

The electrophysiological experiments we report were also never meant to show that the antennal lobe is the “seat” or “locus” of behavioral latent inhibition in the brain. It was not even meant to argue that what happens in the antennal lobe is necessary and/or sufficient for generating behavioral latent inhibition. That, or specifically what roles antennal lobe mechanisms play in behavior, now remains to be determined.

We focus on the antennal lobes simply because we know where amtyr1 is in that circuitry, and we have shown in two previous publications that manifestations of latent inhibition can be seen in the antennal lobe with the recordings. Our intention in putting the electrophysiological data from the antennal lobes into the manuscript was simply to show that we can establish neural correlates of amtyr1 blockade that depend on the same odor familiarization treatment that generates behavioral latent inhibition. These data generate ideas that can be further investigated with more detailed experiments that are beyond the scope of this publication.

I could specifically reference the mechanism described in this summary comment from this reviewer, and hopefully the reviews will be published with the manuscript. But I have to admit I am not sure I understand what is being proposed here well enough to be comfortable putting it in our manuscript, or that it is even fundamentally different from the familiarization-based mechanism that we propose in Figure 7. For example, if familiarization causes “further saturation”, as this reviewer suggests, then by definition the familiarization treatment – plasticity produced by unreinforced odor exposure – is a *correlate* of latent inhibition as defined above. So I don’t understand the phrase “…rather than leading to latent inhibition…”. The effect this reviewer is describing depends on the treatment that produces latent inhibition.

In the end this reviewer may be suggesting something very similar to what we are suggesting – that there is a familiarization-dependent mechanism that affects neural inhibition and hence odor processing in the antennal lobe and, quite likely, mushroom bodies too. Maybe the main difference is that our model in Figure 7 is simplified and meant only as a heuristic to explain what is happening. This reviewer proposes something more specific to the actual neural machinery of the antennal lobe to accomplish the same familiarization-dependent process. But it may not be an independent, alternative interpretation. And it requires a computational model to elaborate on it.

Reviewer #2 (Recommendations for the authors):In the revised version of their manuscript, the authors have responded to many of the reviewers' comments. In particular, they modified the discussion of the data significantly. They introduced a new Figure 7 in order to make it easier for the reader to understand the complex model of the effects of AmTYR1 activation in the antennal lobe.However, I do not find the model very understandable, and, in particular, the results of the electrophysiological recordings shown in Figure 5 are not sufficiently addressed in this model.

See responses to comments 3 and 4 for review #1. The discussion of the (now the correct figure) model has been greatly expanded in lines 501-539 of the discussion.

In an attempt to understand these relations better, I again looked closely at Figure 5C+D of the manuscript. The question I asked myself was what the outcome of a reciprocal experiment would be:How does the number of units (neurons) that are more responsive to octanone change when familiarization is done with octanone instead of hexanol?

In previous study (ref 51 Lei et al.; https://doi.org/10.1371/journal.pone.0265009), also cited in the manuscript and in the new section of the Discussion (lines 519-524), we showed that familiarization shifted antennal lobe units to respond more strongly to novel odor, regardless of which odor – 2-octanone or hexanol – was used as the novel odor. In other words, the odors were fully counterbalanced in that study.

A drop from 39% to a smaller number would likely be expected. Is that true?

Not necessarily. The relative portions above and below the diagonal line depend on collective response biases across all recorded units.

In this case, what is the influence of yohimbine injection?

With latent inhibition intact, more units became biased towards the novel odor after familiarization. With latent inhibition disrupted by yohimbine, the novel odor biases disappeared. We now explain this effect, and its origins, more in the discussion.

If yohimbine prevents the latent inhibition effect, yohimbine injection should prevent a decrease in the number (below 39%) or possibly even cause the number of responding neurons to rise (above 39%). Can this be shown experimentally?

Yes, but it’s actually the opposite. Latent inhibition resulted in percentage increase (from 39% to 54%). Yohimbine caused the percentage to decrease (from 49% to 14%). Again, see new explanation in the Discussion.

Alternatively, does yohimbine lead to a decrease in the number of octanone-biased neurons also in this constellation?

Yes. In this experiment, octanone was a novel odor. Yohimbine caused a decrease of number of units responding to octanone after familiarization.

This would argue for yohimbine causing a general decrease in the response to odorants.

Yes, yohimbine caused a general decrease in response to odorants. As we now explain in the discussion, that is consistent with an expected increase in excitation from ORN axons onto inhibitory interneurons, as shown in Figure 7. However, for either odorant (as in Lei et al. ref 51) one can still identify units that have response biases toward one or the other odor. In the current experiment (Figure 5D), 49% of units showed stronger response to octanone as the novel odor even in the presence of yohimbine. After familiarization under yohimbine treatment, response bias did not increase as it normally would. Instead it decreased such that only 14% of units remained more responsive to octanone. In other words, familiarization normally causes more units to respond to novel odor, but that was disrupted by yohimbine.

Can these assumptions or the results of the corresponding experiment be reconciled with the model shown in Figure 7? What would the model imply in this case?

Biases toward one or the other odor prior to familiarization are consistent with many other studies of units in the antennal lobes of many insects. And these biases occur even after yohimbine treatment, which causes a general decline in unit responsiveness. But the increase in response to the novel odor after familiarization is blocked, and even reversed, by yohimbine.

There are many open questions that we can speculate on, but it would be better done in the context of more experiments with different odors, more physiological recordings both from the antennal lobes and mushroom bodies, and, importantly, all of this should be done in the context of rigorous computational modeling.

We strongly agree with the comment from this reviewer in the first review. That is, that we should adopt the fruit fly to further test what we propose here. And we are in the process of doing that.

Line 482 "this plasticity it would give rise to": Delete "it".

Deleted.